# Dormant pathogenic CD4+ T cells are prevalent in the peripheral repertoire of healthy mice

Anna Cebula [1], Michal Kuczma [1], Edyta Szurek[1], Maciej Pietrzak [2], Natasha Savage[3], Wessam R. Elhefnawy[4], Grzegorz Rempala [2], Piotr Kraj [5] & Leszek Ignatowicz [1]*

Thymic central tolerance eliminates most immature T cells with autoreactive T cell receptors (TCR) that recognize self MHC/peptide complexes. Regardless, an unknown number of autoreactive CD4+Foxp3− T cells escape negative selection and in the periphery require continuous suppression by CD4+Foxp3+ regulatory cells (Tregs). Here, we compare immune repertoires of Treg-deficient and Treg-sufficient mice to find Tregs continuously constraining one-third of mature CD4+Foxp3− cells from converting to pathogenic effectors in healthy mice. These dormant pathogenic clones frequently express TCRs activatable by ubiquitous autoantigens presented by class II MHCs on conventional dendritic cells, including self-peptides that select them in the thymus. Our data thus suggest that identification of most potentially autoreactive CD4+ T cells in the peripheral repertoire is critical to harness or redirect these cells for therapeutic advantage.

[1] Institute for Biomedical Sciences, Georgia State University, Atlanta, GA, USA. [2] Mathematical Biosciences Institute, Ohio State University, Columbus, OH, USA. [3] Department of Pathology, Augusta University, Augusta, GA, USA. [4] Department of Computer Science, Old Dominion University, Norfolk, VA, USA. [5] Department of Biological Sciences, Old Dominion University, Norfolk, VA, USA. *email: lignatowicz@gsu.edu

The current paradigm proposes that in healthy individuals, peripheral CD4[+] T cells rarely become autoreactive because most thymocytes expressing αβTCRs (TCRs) which bind self MHC/peptide complexes with moderate to high affinity are eliminated by thymic negative selection[1–4]. Alternatively, thymocytes that express potentially autoreactive TCRs are directed to CD4[+]Foxp3[+] or CD4[+]CD8αα[+] T cell lineages or rendered unresponsive by mechanisms of peripheral tolerance[5]. In contrast, several reports investigating the effectiveness of negative selection by different model autoantigens concluded that a significant portion of autoreactive CD4[+] cells escape to the periphery, although pathogenicity of these cells in vivo has not been examined[6,7]. The frequency of self-reactive CD4[+] cells appears to be inversely proportional to the abundance of the relevant selecting autoantigen in the thymus[8], but global identification and quantification of most self-reactive, mature CD4[+] cells present in the peripheral repertoire remains a challenge.

Finding potentially autoreactive CD4[+] cells in healthy individuals is also a challenging task because mechanisms of peripheral tolerance including anergy, expression of inhibitory co-receptors and suppression mediated by Tregs continuously sustain quiescence of self-reactive T cells. Thus, it is not surprising that blockade of inhibitory receptors on T cells or depletion of Tregs precipitate autoimmunity[9,10], revealing a potential risk of clinical approaches which, in order to improve immune response to infection or cancer, attempt to weaken peripheral tolerance mechanisms[11,12]. Acute course of fatal autoimmunity in mice and humans with congenital Foxp3 defects also implies that an abundance of autoreactive CD4[+]Foxp3[−] cells and diversity of their TCRs may need to be re-examined to better understand the pathogenesis of autoimmune diseases.

Several studies have attempted to identify CD4[+] cells expressing potentially autoreactive TCRs in healthy mice and to reveal their self-reactivities by compromising peripheral tolerance. Experiments in which Tregs were deleted using aCD25 MoAb or diphtheria toxin (DT) concluded that the proportion of autoreactive CD4[+] clones in the normal repertoire detected by an increase in expression of CD69, Nur77[GFP] or measuring expansion of autoreactive T cell clones is around 4%[13–16]. In other reports, the ablation of Tregs by DT in specific pathogen-free (SPF) or germ-free adult mice led to rapid onset of lethal autoimmunity, demonstrating that many peripheral CD4[+]Foxp3[−] cells can be activated by tissue-derived autoantigens unless these cells are stopped by Tregs[17,18].

Here we report that in adult, healthy C57BL/6 mice one-third of CD4[+]Foxp3[−] T cells express autoreactive TCRs that can be activated by ubiquitous self-peptides presented by autologous DCs. Normally, these cells pathogenic autoreactivity is constrained by Tregs, but when Tregs are dysfunctional or ablated it changes CD4[+]Foxp3[−]cells activation threshold for self-peptides. We propose that common selection by agonist self-peptides and limited intrathymic deletion enrich peripheral repertoire of CD4[+]Foxp3[−] cells in many dormant autoreactive clones that in suitable conditions can cause autoimmunity.

## Results

### Scurfy (Sf) mice with restricted αβTCR repertoire succumb to autoimmunity.
Sf mutant mice lack functional Tregs and develop lethal autoimmunity driven by polyclonally activated CD4[+] cells[19]. The identities of autoreactive CD4[+] clones exclusive to the Sf strain versus those shared with healthy mice have not been investigated. This analysis can be achieved by comparing autoreactive TCRs on peripheral CD4[+]Foxp3[−] cells in Sf mice with TCRs expressed by cells with activated phenotype present in healthy mice. To ensure that we could analyze cells expressing the same set of TCRs, we used TCR[mini] mice where TCRs diversity is reduced approximately by two orders of magnitude (from $10^7$ of different TCRs in wild type mice[20] to ~$10^5$ in our strains).To compare TCR repertoires, we first backcrossed SfC57BL/6 to TCRα[−] mutant mice, and then to TCR transgenic mice expressing a single, rearranged Vβ14Jβ2.6 chain paired with different, transgenic TCRα chains formed by natural rearrangements of single Vα2.9 segment to one of the two Jα segments (Jα2/Jα26). In these mice, a semi-diverse TCR repertoire supports natural thymocyte differentiation to CD8[+], CD4[+]Foxp3[−] and Foxp3[+] lineages and CD4[+] cells respond to various antigens[21]. We further crossed SfTCR[mini] mice with C57BL/6Foxp3[GFP] (now referred to as SfTCR[mini] or TCR[mini] mice) or C57BL/6Nur77[GFP] reporter strains, in which GFP expression labels Tregs or reflects TCR functional affinity for self-antigen(s), respectively[22,23]. In the SfTCR[mini]Foxp3[GFP] model, dysfunctional Tregs can be separated from activated, CD4[+]Foxp3[GFP−] cells; in SfTCR[mini]Nur77[GFP] mice, autoimmune effectors can be categorized according to their TCR signal strength.

Figure 1a shows that SfTCR[mini] mice develop lethal autoimmunity by 7–10 weeks of age, and 6-week-old SfTCR[mini] mice suffer from lymphadenopathy, splenomegaly, and numerous infiltrates of CD4[+] cells in the lungs, liver, and skin (Fig. 1b). In various organs isolated from SfTCR[mini] mice, 65–85% of CD4[+]Foxp3[GFP−] cells (SfFoxp3[GFP−]) showed an activated CD44[+]CD62L[−] phenotype and elevated expression of the proliferation marker CD71 (Fig. 1c, d, Supplementary Fig. 1a). Overall, polyclonal activation of CD4[+] cells in SfTCR[mini] mice increased the total number of CD4[+] cells compared to TCR[mini] controls (Fig. 1e), whereas the proportions of CD4[+] cells with reporter Foxp3[GFP+] (SfFoxp3[GFP+]) resembled proportions of CD4[+]Foxp3[GFP+] cells in healthy mice (with the exception of colon where their proportion was lower) (Fig. 1f, g). Ultimately, all SfTCR[mini] mice developed clinical symptoms of fulminating autoimmune syndrome manifested by runting, exfoliative dermatitis and wasting disease. Together these results show that SfTCR[mini] mice develop multi-organ autoimmunity that resembles disease in original SfC57BL/6 mice and that TCR[mini] mice need functional Tregs to sustain self-tolerance.

### CD4[+]Foxp3[−] cells from Sf and healthy mice share many TCRs.
The relationship between autoreactive CD4[+] clones in Sf mice and peripheral CD4[+] cells in healthy mice remains unknown. This is in part because thymic atrophy in Sf mice supports the release of self-reactive CD4[+] cells, and proinflammatory cytokines expand rare autoreactive clones[24]. Alternatively and not mutually exclusive, when Tregs are disabled, a substantial portion of naive CD4[+]Foxp3[GFP−] cells (which are also present in healthy mice) become activated by self-antigens[17]. We examined TCRs expressed by activated CD4[+] clones in SfTCR[mini] mice and compared them to TCRs on CD4[+] cells from healthy TCR[mini] mice. For this purpose, we used high throughput sequencing (HTS) that examined over $4 \times 10^6$ of TCRα CDR3 regions from each TCR[mini] strain studied. This approach showed that most of dominant TCRα CDR3 regions retrieved from SfFoxp3[GFP−] effectors (CD4[+]CD44[+]CD62L[−]) were also found on Foxp3[GFP−] clones in healthy mice (Fig. 2a, Supplementary Fig. 2), and the shared portion was similar when respective repertoires of CD4[+]Foxp3[GFP+] cells were compared (Fig. 2b, Supplementary Fig. 2). The substantial overlap of TCRs expressed by CD4[+] cells from SfTCR[mini] and TCR[mini] mice was also confirmed using single-cell TCR sequencing (Supplementary Fig. 3). To examine the global similarity between these repertoires, we applied the Mutual Information Index (MII) which assigns a value of 1 to a pair of completely overlapping repertoires (containing identical TCRs abundance patterns) and value of 0 to

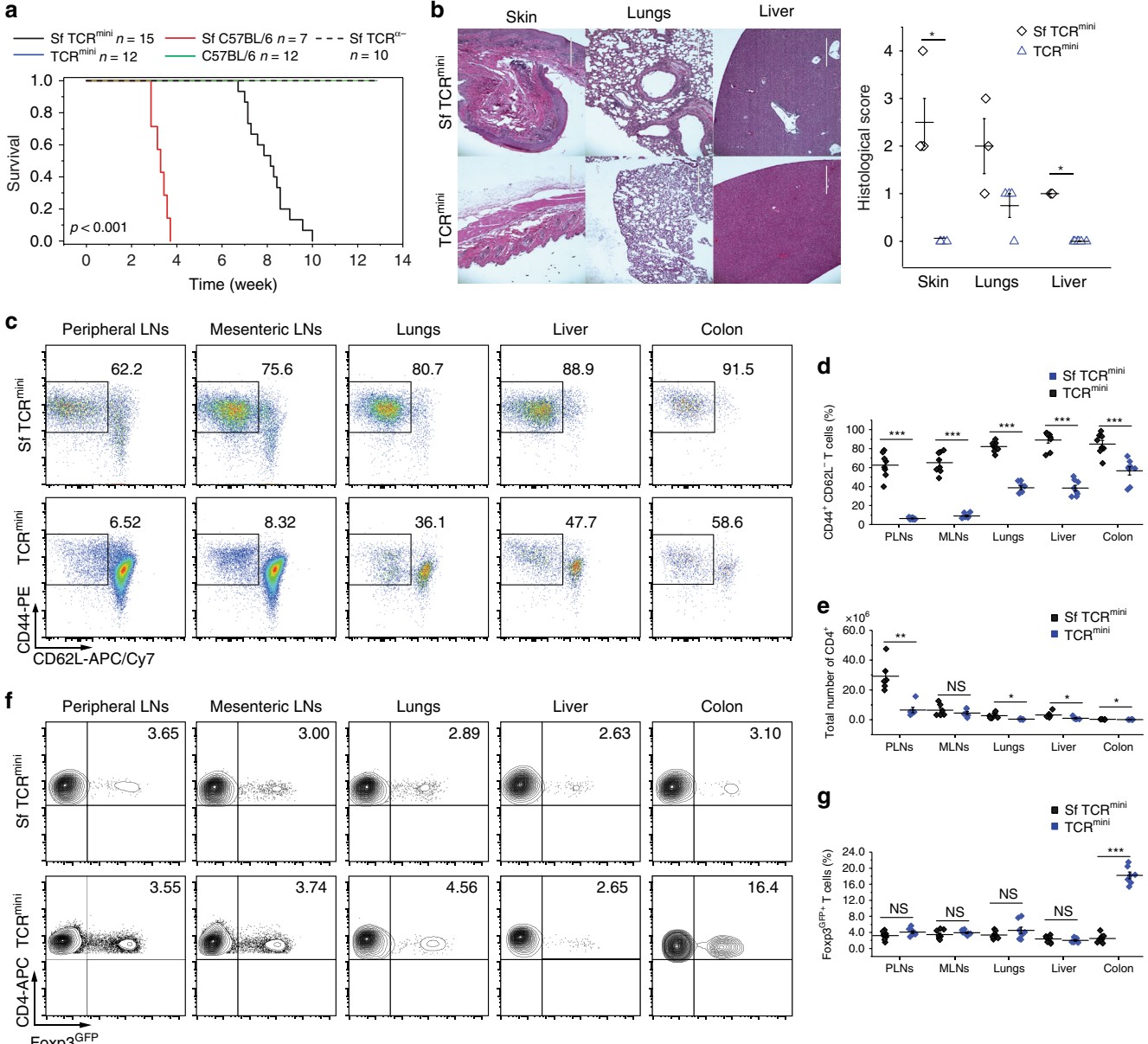

**Fig. 1** SfTCR[mini] mice develop multiorgan disease as original Sf mice. **a** Survival curves of SfC57BL/6 and SfTCR[mini] mice and respective healthy strains (Kaplan–Meier curves and log-rank test; $n = 7$–16 per group). **b** Photomicrographs of H&E stained histological sections of various organs from SfTCR[mini] (upper panel) and TCR[mini] mice (lower panel). For each strain, 3–6 mice were examined and each symbol represents the individual animal. Scale bar 400 μm. **c**, **d** Proportions of CD4[+]Foxp3[GFP−]CD44[+]CD62L[−] cells in lymphoid and non-lymphoid organs. **e** Total number of CD4[+] cells and **f**, **g** proportions of CD4[+]Foxp3[GFP+] cells in indicated organs. All mice were 6–8 weeks old, each dot on graphs represents one mouse ($n = 5$–10). Unpaired $t$-test was applied, and statistical significance is indicated where appropriate (*$p < 0.05$, **$p < 0.01$, ***$p < 0.001$, NS-not significant). Mean ± SD is shown for each sample. Source data is provided in a Source Data file

pairs with entirely non-overlapping patterns[25]. The values of this index for repertoire of CD4[+]Foxp3[GFP−] subsets ranged from 0.3 to 0.7, demonstrating a significant overlap of both repertoires (Fig. 2c). Approximately 25% TCRs were not found in the TCR[mini] repertoire, suggesting that Tregs-deficiency also supports an expansion of strain-specific clones. To compare the diversity of TCRs on CD4[+] cells from Sf and control TCR[mini] mice, we utilized the Renyi Entropy Function (REF). REF analyzes the diversity of TCRs considering unseen species and adjusts for the abundance of specific TCRs. A REF value of <1 gives more weight to the rare TCRs contributing to the repertoire, whereas a REF value of greater than 1 gives more weight to the abundant TCRs[25]. The REF calculated here shows that the overall SfCD4[+] repertoire was more

heterogeneous than the corresponding repertoire in healthy mice likely due to an influx of low-abundance clones (Fig. 2d).

**SfCD4[+] cells exhibit exaggerated reactions to self-antigens.** Since many CD4[+]Foxp3[GFP−] cells from TCR[mini] and SfTCR[mini] mice expressed mutual TCRs but only the latter strain developed autoimmunity, we investigated ex vivo autoreactivity of individual TCRs. For this purpose, we sorted activated CD4[+]Foxp3[GFP−] cells from both strains, produced CD4[+]TCR[+] hybridomas and studied their activation by bone marrow-derived dendritic cells (DCs) from C57BL/6 mice. We measured TCR driven hybridomas activation by examining IL-2 production and fold increase over the background in Nur77[GFP] expression[23]. Unstimulated

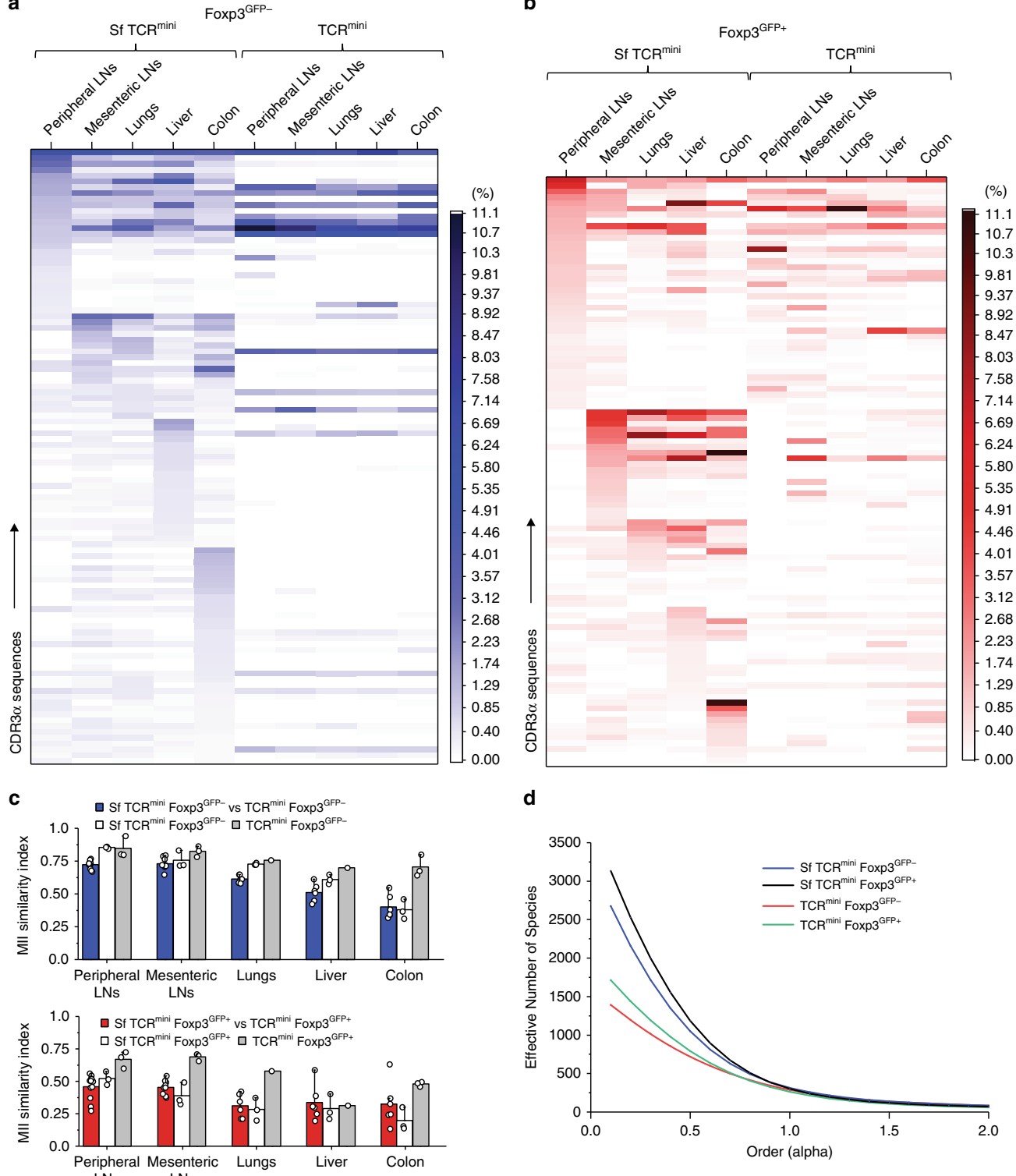

**Fig. 2** CD4[+] cells from SfTCR[mini] and TCR[mini] mice share many αβTCRs. **a** Frequencies of 50 dominant TCRs on CD4[+]Foxp3[GFP−] (blue heatmap) or **b** Foxp3[GFP+] (red heatmap) cells from each indicated organ (sequences see Supplementary Data 1a, b). **c** Similarity indices (MII) of TCRs expressed by CD4[+]Foxp3[GFP−] (upper panel) or CD4[+]Foxp3[GFP+] (lower panel) cells among individual SfTCR[mini] (white bars) or TCR[mini] (gray bars) mice or cross-compared among both strains (blue and red bars). **d** Diversity (REF) calculated for repertoires of CD4[+]Foxp3[GFP−] (or Foxp3[GFP+]) cells from Sf or control mice. Repertoires from three individual mice from each strain were sequenced. Source data are provided in a Source Data file. Error bars depict SD

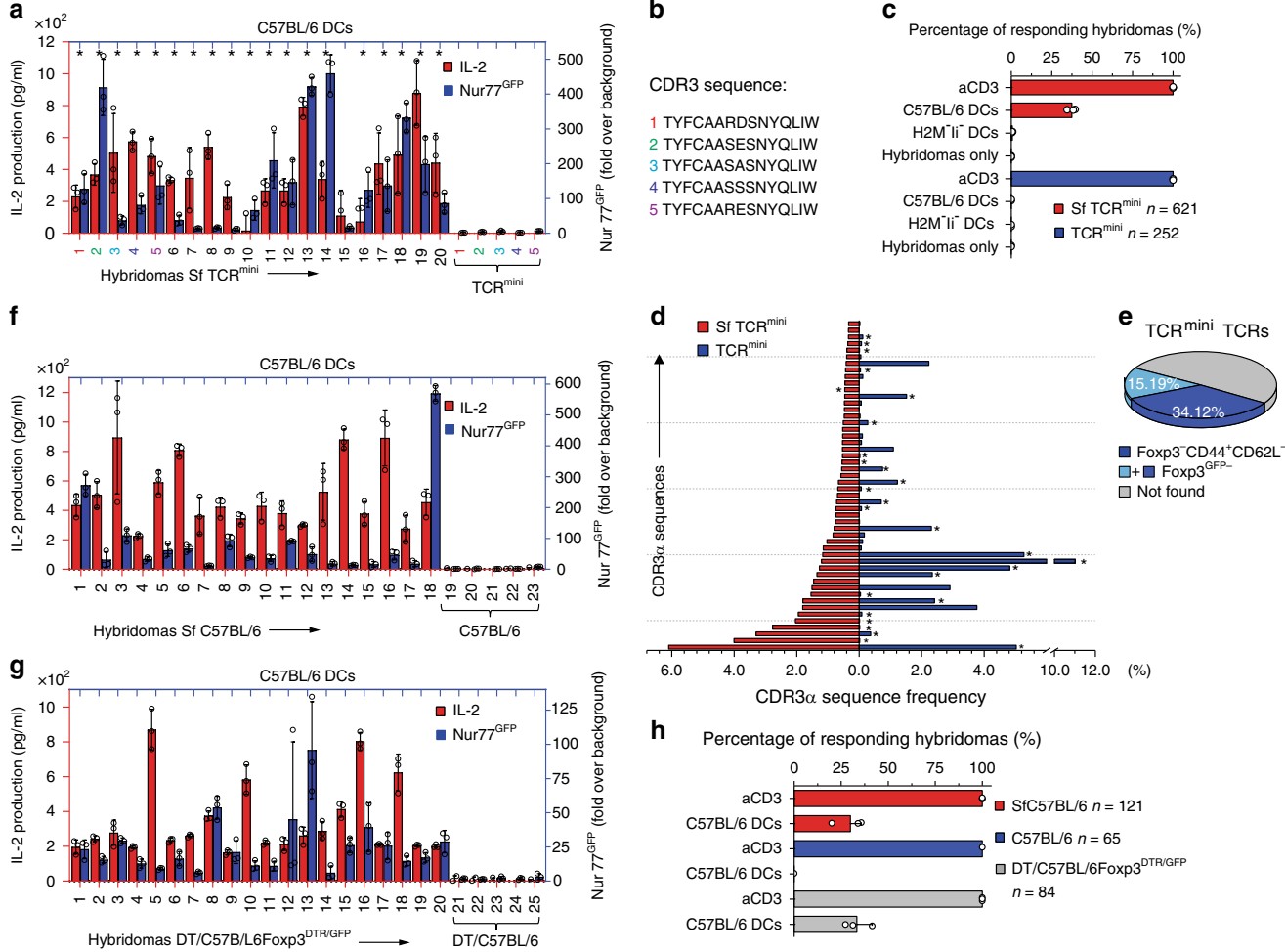

**Fig. 3** Hybridomas from SfCD4+Foxp3GFP− cells are activated by C57BL/6 DCs. **a** IL-2 production (red bars) and Nur77GFP expression (navy bars) by representative hybridomas (#1-20) from SfTCRmini mice. Hybridomas were stimulated overnight by autologous DCs. The last 5 hybridomas (marked by colored #1-5) were established from TCRmini mice and expressed the same five TCRs as first hybridomas #1-5 from SfTCRmini mice. Stars mark hybridomas whose TCRs are also expressed by naive CD4+Foxp3GFP− cells in healthy TCRmini mice. **b** Sequences of TCRα CDR3 regions from hybridomas marked # 1-5. For other TCRα CDR3 sequences see Supplementary Data 5. **c** One-third of hybridomas derived from CD4+Foxp3GFP− SfTCRmini effectors recognize peptides presented by Abwt, but not empty Ab on DCs from H2-M−Ii− mice. Bars show % of responders of the total number of tested hybridomas from each strain. **d** Top fifty dominant TCRs expressed by Sf effectors (CD4+CD44+CD62L−Foxp3GFP−) in LNs, and these TCRs frequencies on CD4+Foxp3GFP− cells in the same organ in TCRmini mice (for sequences see Supplementary Data 6). Stars (*) mark TCRs also expressed by SfCD4+ cell hybridomas that responded ex vivo to autologous DCs. **e** TCRs retrieved from autoreactive SfCD4+Foxp3GFP− hybridomas account for a major fraction of TCRs expressed by peripheral CD4+Foxp3GFP− cells from healthy TCRmini mice. **f** IL-2 production and Nur77GFP expression by representative hybridomas from SfC57BL/6 mice (#1–18) or healthy C57BL/6 mice (#19-23). Hybridomas were activated as indicated in (a). **g** IL-2 production and Nur77GFP expression by representative CD4+Foxp3GFP− hybridomas from toxin-treated C57BL/6 Foxp3DTR/GFP(#1-20) or DT-treated control mice (#21-25) after o/n co-culture with autologous DCs. Error bars are standard error across 3 SD replicas. **h** Percent of all tested hybridomas from Sf and control C57BL/6 or toxin treated C57BL/6 Foxp3DTR/GFP that responded to activation by autologous DCs. Bars represent % of responders of total number of tested hybridomas from each strain. Source data are provided in a Source Data file. Error bars shown as SD

hybridomas (621 from Sf and 252 from control CD4+Foxp3GFP− cells) did not constitutively make IL-2, had comparable expression of CD4 or TCR and expressed only a basal level of the Nur77GFP reporter (Supplementary Fig. 4). However, all hybridomas reproducibly made IL-2 and expressed GFP when incubated overnight on aCD3 MoAb coated plastic plates (Supplementary Fig. 5a). Significantly, ~37% of the hybridomas established from SfCD4+Foxp3GFP− cells produced IL-2 and elevated Nur77GFP expression following overnight co-culture with C57BL/6 DCs, whereas less than 0.5% of hybridomas from healthy TCRmini mice showed similar autoreactivity (Fig. 3a, c, Supplementary Fig. 6a). Although most of SfCD4+Foxp3GFP− derived hybridomas augmented PD-1 (but not CTLA-4) expression when stimulated by C57BL/6 DCs, this change could not

prevent these cells activation (Supplementary Fig. 7). SfCD4+Foxp3GFP− derived hybridomas indiscriminately responded to bone marrow or splenic DCs from SfC57BL/6 or C57BL/6 donors (Supplementary Fig. 8a), and these responses did not change after DCs were preloaded overnight with two organ-specific lysates (Supplementary Fig. 8b), which suggests that most activating self-peptides are naturally presented by DCs from healthy mice. In contrast, <1% of SfCD4+Foxp3GFP− hybridomas responded to DCs from C57BL/6 H2M−Ii− mice (Fig. 3c), which due to impaired antigen processing express normal levels but mostly empty class II MHC molecules (Ab)[26]. This last result demonstrates that these hybridomas activation is triggered by autoantigens bound to Ab. We also found that hybridomas generated from SfCD4+TCR+ thymocytes did not respond ex vivo to

self-peptides presented in the same conditions (Supplementary Fig. 8c–e). In addition, 29% of hybridomas established from SfCD4+Foxp3GFP+ cells become activated following overnight co-culture with C57BL/6 DCs (Supplementary Fig. 9), demonstrating that abrogating Foxp3 function reveals comparable proportions of autoreactive clones in both CD4+ subsets.

Next, we sequenced TCRs expressed by 232 autoreactive hybridomas from SfCD4+Foxp3GFP− cells (Supplementary Data 5) and compared their CDR3α regions to relevant repertoires from SfTCRmini and TCRmini mice. First, we found that autoreactive hybridomas expressed more than half of 50 most abundant TCRs originally sequenced from SfCD4+Foxp3GFP− cells and that many of the same TCRs were also expressed by CD4+Foxp3GFP− cells from TCRmini mice (marked by an asterisk in Fig. 3d). Next, we compared all CDR3α sequences from autoreactive hybridomas from SfCD4+Foxp3GFP− cells to complete databases of CDR3α regions retrieved from TCRmini repertoire of CD4+Foxp3GFP− cells. As shown in Fig. 3e, we found that autoreactive TCRs accounted for 34% of CDR3α regions sequenced from activated CD4+Foxp3GFP− cells (overall 49% of all TCRs sequenced from CD4+Foxp3GFP− cells), confirming that potentially pathogenic clones constitute a substantial fraction of the TCR repertoire in normal, healthy TCRmini mice. We also sequenced TCRs expressed by 252 self-tolerant hybridomas from CD4+Foxp3GFP− cells and discovered TCRs that were identical with previously sequenced autoreactive TCRs from SfCD4+Foxp3GFP− hybridomas (examples in Fig. 3a, b marked #1-5). Even though these shared TCRs had matching specificities for self-peptides, only Sf-derived hybridomas responded ex vivo to C57BL/6 DCs (Fig. 3a).

Finally, we identified three TCRs in a set of SfCD4+Foxp3GFP− autoreactive hybridomas that matched TCRs on hybridomas generated from prostate draining lymph node CD4+ T cells isolated from TCRminiAire− mice. Aire− males develop non-lethal prostatitis because defective presentation of Aire−controlled self-peptides impairs negative selection in the thymus, decreases elimination of autoreactive CD4+ effectors and restricts development of tissue−specific Tregs[27]. In fact, CD4+Foxp3GFP− hybridomas generated from TCRminiAire− mice responded only to C57BL/6 DCs preloaded with prostate autoantigens (Supplementary Fig. 10). These findings suggest that broad or tissue-specific deficits of Tregs cells in Sf and Aire- mice impact activation of CD4+ cells expressing identical TCRs specific for ubiquitous and organ-specific autoantigens.

**In C57BL/6 mice Tregs conceal many autoreactive CD4+ Foxp3− cells.** In TCRmini mice we found that CD4+ cells from autoimmune-prone Sf and healthy control mice (Fig. 3c) share around 30% of autoreactive TCRs, but the relevance of this observation to mice expressing wild type TCR repertoire remained undefined. Therefore, we produced hybridomas representing CD4+Foxp3GFP− cells from SfC57BL/6 (or control C57BL/6) mice and examined these hybridomas self-reactivity to autologous DCs. As shown in Fig. 3f, h ~29% of hybridomas established from SfC57BL/6 mice responded to C57BL/6 DCs, whereas >1% of hybridomas derived from control C57BL/6 mice showed detectable activation, though all tested hybridomas similarly responded to activation by immobilized αCD3 MoAb (Supplementary Figs. 5b and 6b). Thus, the abundance of autoreactive CD4+Foxp3GFP− clones is not artificially enhanced in TCRmini mice as a result of the narrower repertoire. However, the percentage of autoreactive CD4+Foxp3GFP− cells from SfC57BL/6 mice could not be used to approximate the size of a respective subset in healthy C57BL/6 mice because thymic involution and duration of the disease in Sf mice supports differentiation and clonal expansions of strain-specific, autoreactive cells. In addition,

cross-comparison of a representative number of autoreactive TCRs shared by CD4+Foxp3− cells from SfC57BL/6 and C57BL/6 will be cost-prohibitive.

To overcome these obstacles, we also immortalized CD4+ Foxp3GFP−cells from adult C57BL/6Foxp3DTR/GFP mice that for 5 consecutive days received diphtheria toxin injections, which resulted in complete Tregs deletion and fatal autoimmunity on day 6 or 7. In toxin injected C57BL/6Foxp3DTR/GFP mice over 80% of residual CD4+Foxp3GFP− cells had activated phenotype, which was not observed in similarly treated control mice[18]. As shown in Fig. 3g, h, Supplementary Figs. 5c and 6c, 33% of hybridomas representing CD4+Foxp3GFP− from toxin-treated C57BL/6Foxp3DTR/GFP mice were activated after overnight co-culture with autologous DCs, demonstrating that repertoire of C57BL/6 mice contains a significant portion of dormant, autoreactive cells.

Recently it has been shown that pathogenicity of autoreactive CD4+Foxp3GFP− cells in adult DT-treated C57BL/6Foxp3DTR/GFP mice can be attenuated by an injection of pharmacological inhibitor GSK503 that suppresses histone lysine methyltransferase Ezh2, a key enzymatic component of cytosolic polycomb repressive complex 2 (PRC2)[28]. PRCP2 modulates TCR signaling by interfering with TCR-mediated activation of MAPK/Erk and IL2/IL2Rα expression. To determine if autoreactivity of CD4+ Foxp3GFP− hybridomas established from Tregs deficient mice may reflect differences in epigenetic mechanisms regulating TCR signaling, we pretreated these and control hybridomas with Ezh2 inhibitor prior to their stimulation by C57BL/6 DCs. As shown in Supplementary Fig. 11, pretreatment with 5 μM GSK503 attenuated TCR-driven responses of autoreactive hybridomas to autologous DCs but had no effect on these cells activation by stimulants that bypass TCR/CD3 triggering. Therefore, Tregs induced changes in epigenetic mechanisms that control TCR activation in CD4+Foxp3− T cells can be reproduced ex vivo in hybridomas derived from these cells.

**Sf mice with A^b bound by single peptide succumb to autoimmunity.** Significant overlap between autoreactive TCRs expressed by CD4+ cells in SfTCRmini and TCRmini mice can result from intrinsic limitations of negative selection in the thymus. Negative selection targets thymocytes expressing such TCRs, but the highly diverse repertoires of TCRs and confronted self-antigens impose spatial and temporal limits on the effectiveness of deletion[8,29,30]. To improve the effectiveness of negative selection, we generated SfTCRmini mice that express class II MHC (A^b) molecules covalently bound with a single autoantigen (Eα) and Foxp3GFP reporter (now referred to as SfTCRminiA-bEp). The diversity of natural self-peptides bound to A^b is eliminated in these mice, whereas A^bEp covalent complexes naturally select and sustain diverse repertoires of CD4+Foxp3+ and Foxp3− cells[31]. Unexpectedly, while TCRminiA^bEp mice have a normal life expectancy[31], SfTCRminiA^bEp mice developed rampant autoimmunity within the first weeks of life and had to be euthanized at 6 weeks of age, significantly earlier than SfTCRmini mice (Figs. 1a, 4a). As shown in Fig. 4b, SfTCRminiA^bEp had severe infiltrates in their skin and liver, the higher total number of activated and proliferating CD4+ effectors (Fig. 4c–e, and Supplementary Fig. 1b) but fewer SfCD4+ Foxp3GFP+ cells (Fig. 4f, g). Thus, in SfTCRminiA^bEp autoimmune manifestations affected the same organs as in the original Sf mice despite that the former strain lacks expression of tissue-specific antigens. Instead, the majority of autoreactive SfCD4+Foxp3GFP− cells recognized the covalent A^bEp complexes as their targets, as SfTCRminiA^b− mice lived much longer before succumbing to the disease (Fig. 4a). In this latter strain,

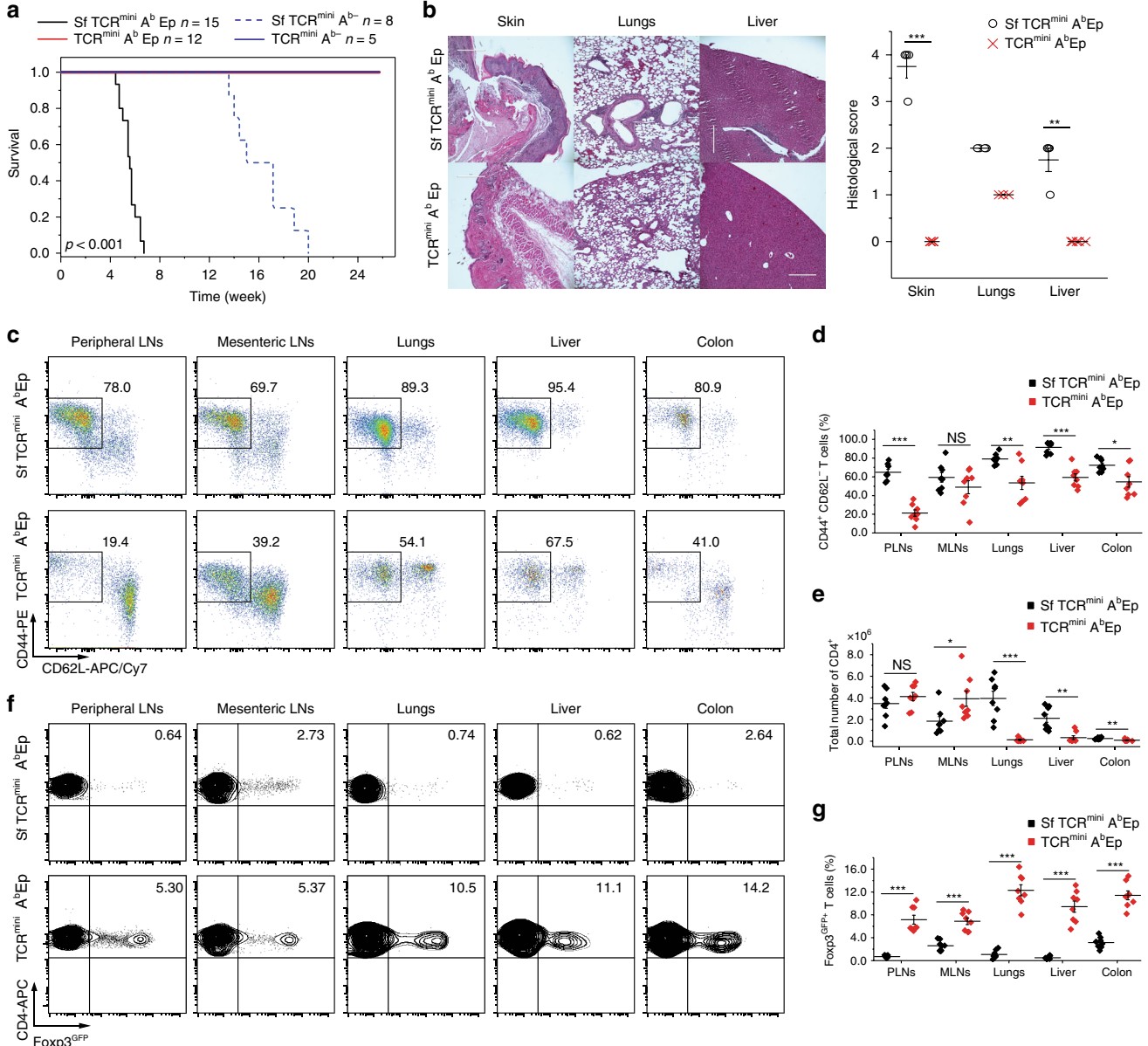

**Fig. 4** Single peptide SfTCR^mini A^b Ep mice succumb to autoimmunity. **a** Survival curves of SfTCR^mini A^b Ep, SfTCR^mini A^b- and corresponding TCR^mini strains. **b** Photomicrographs of H&E stained histological sections of organs from TCR^mini A^b Ep (lower panel) and SfTCR^mini A^b Ep mice (upper panel). For details see Fig. 1b. Scale bar 400 μm. **c**, **d** Proportions of CD4+CD44+CD62L− cells in indicated mice. **e** The total number of CD4+ cells and **f**, **g** Proportions of CD4+Foxp3^GFP+ cells in indicated organs. All mice were 4-6 weeks old ($n = 5$-10). Unpaired $t$-test was applied, and statistical significance is indicated where appropriate (*$p < 0.05$, **$p < 0.01$, ***$p < 0.001$, NS-not significant). Source data are provided in a Source Data file. Error bars show SD

only a few SfCD4+ cells remained that underwent a selection on non-classical MHC molecules but after 6–8 months these cells expanded causing fatal autoimmunity.

An analysis of the TCR repertoire from SfTCR^mini A^b Ep mice using HTS (Supplementary Fig. 12a, b) showed that 60% of dominant TCRs found on activated SfCD4+ effectors were also found on CD4+ cells in lymph nodes of healthy TCR^mini A^b Ep mice. This high similarity was also indicated by the MII indexes which showed overlaps of 40–65% between TCRs (Supplementary Fig. 12c). The overall diversity of TCRs on CD4+Foxp3^GFP− cells from SfTCR^mini A^b Ep strain was higher than on corresponding cells from control mice, due to increased contribution of rare autoreactive clones (Supplementary Fig. 12d). Taken together, these results show that intrathymic expression of single self-peptide does not eliminate the need for Tregs-induced tolerance for the same class II MHC/peptide complexes expressed in the periphery. Instead and rather counterintuitively, it accelerated the onset and progression of autoimmunity, suggesting that the selecting autoantigen is recognized by self-selected CD4+ cells in the periphery as a cognate autoantigen (see below).

The self-peptide recognized by autoreactive cells in SfTCR^mini A^b Ep mice is known because these mice express only covalently linked A^b Ep complexes. To show directly that selecting self-peptides can activate peripheral SfCD4+Foxp3^GFP− cells and drive autoimmunity, we converted these cells to hybridomas. One-third of tested hybridomas responded in vitro to DCs presenting only A^b Ep complexes, whereas only a few of these hybridomas responded to DCs expressing A^b bound by other, covalently linked A^b Ep63K peptide (Fig. 5a, b). This demonstrates that self-reactive cells are highly specific for the original A^b Ep autoantigen. Also, half of hybridomas that responded to DCs expressing other A^b Ep63K complexes also responded to A^b− DCs, suggesting that

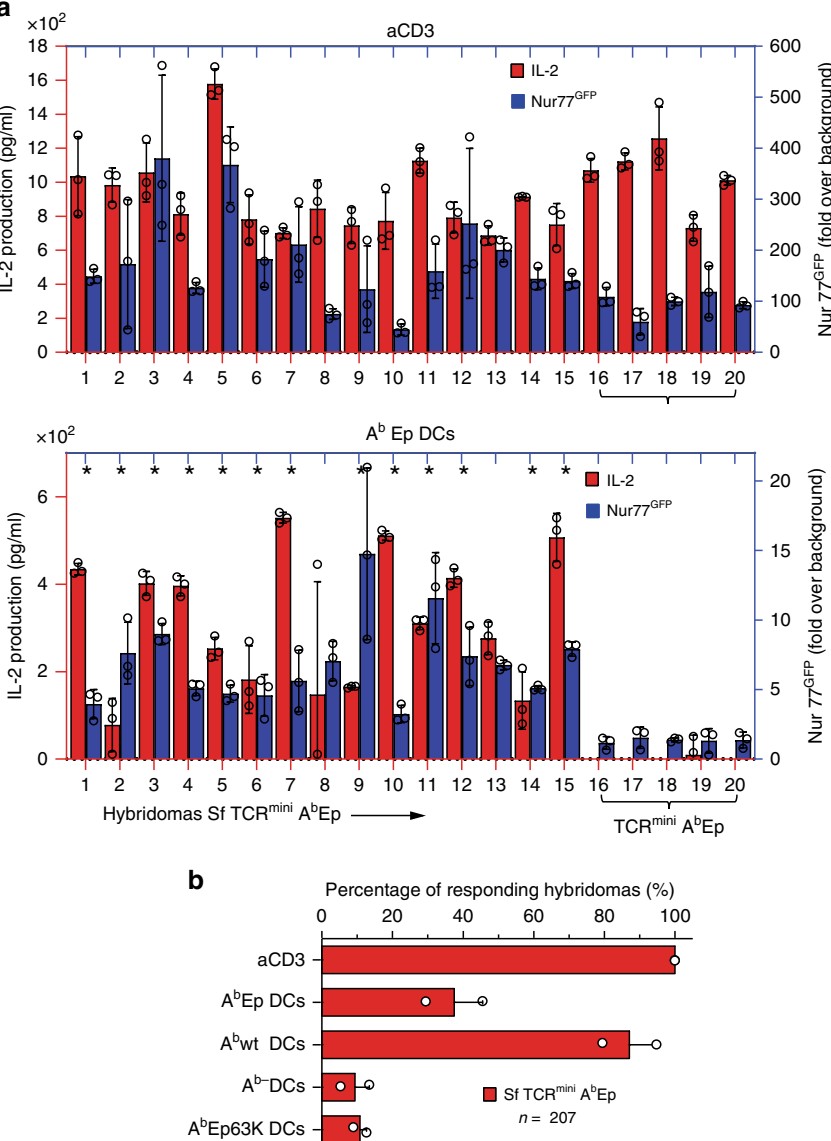

**Fig. 5** Positively selecting self-peptide activates SfCD4+ hybridomas. **a** IL-2 production (red bars) and Nur77GFP reporter expression (navy bars) by representative hybridomas from SfTCRminiAbEp (#1-15) or control TCRminiAbEp (#16-20) mice stimulated by aCD3 MoAb (upper panel) or autologous DCs expressing only AbEp complexes (lower panel). For TCRα CDR3 sequences see Supplementary Data 9. Stars mark hybridomas whose TCRs are also expressed by naive CD4+Foxp3GFP− cells in healthy TCRminiAbEp mice. **b** Percentage of hybridomas from SfTCRminiAbEp mice that responded to DCs expressing endogenous self-peptides (Abwt), covalently linked autologous (AbEp) or non-autologous (AbEp63K) self-peptide, or lacking Ab expression (Ab−). Source data are provided in a Source Data file. Error bars show SD

their TCRs cross-recognize non-classical MHC/autoantigen motifs (Fig. 5b)[31]. As expected, >80% of CD4+ hybridomas from both SfTCRminiAbEp and control TCRminiAbEp mice responded to DCs isolated from C57BL/6 mice, because CD4+ cells isolated from mice expressing a single type of Ab/peptide complexes are not tolerant to other, endogenously processed self-antigens[31]. In summary, these results show that Tregs are indispensable for the induction of peripheral tolerance to autoantigen(s) recognized by CD4+ cells, regardless of how abundantly the autoantigen is expressed during thymic selection.

To investigate the mechanism behind accelerated onset and progression of autoimmunity in SfTCRminiAbEp mice we hypothesized that when the same class II MHC/peptide complexes are abundant in the thymus and the periphery, more autoreactive CD4+ cells require suppression by Tregs. To test this hypothesis, we reconstituted lethally irradiated lymphopenic

AbwtTCRα−, AbEp63KTCRα− and AbEpTCRα− recipients with T-cell depleted bone marrow from SfTCRminiAbEp mice. We found that chimeras expressing AbEp complexes on thymic epithelial and peripheral antigen-presenting cells succumbed to autoimmunity first and had to be euthanized no later than 7 weeks after reconstitution. In contrast, recipients that originally expressed only Ab bound with other covalently linked (AbEp63K) or many self-peptides had the onset of autoimmunity delayed by one to four weeks (Fig. 6a). The SfTCRminiAbEp to AbEpTCRα− chimeras had a high proportion but low total number of CD4+ Foxp3GFP− effectors. Their TCRs resembled those expressed from the corresponding subset in SfTCRminiAbEp mice, suggesting that pathogenicity in both mice involves the same CD4+ clones (Fig. 6b–d). This pathogenic repertoire had the lowest diversity of TCRs of all compared chimeras, suggesting that these CD4+ cells underwent the most rapid clonal expansions (Fig. 6e). In

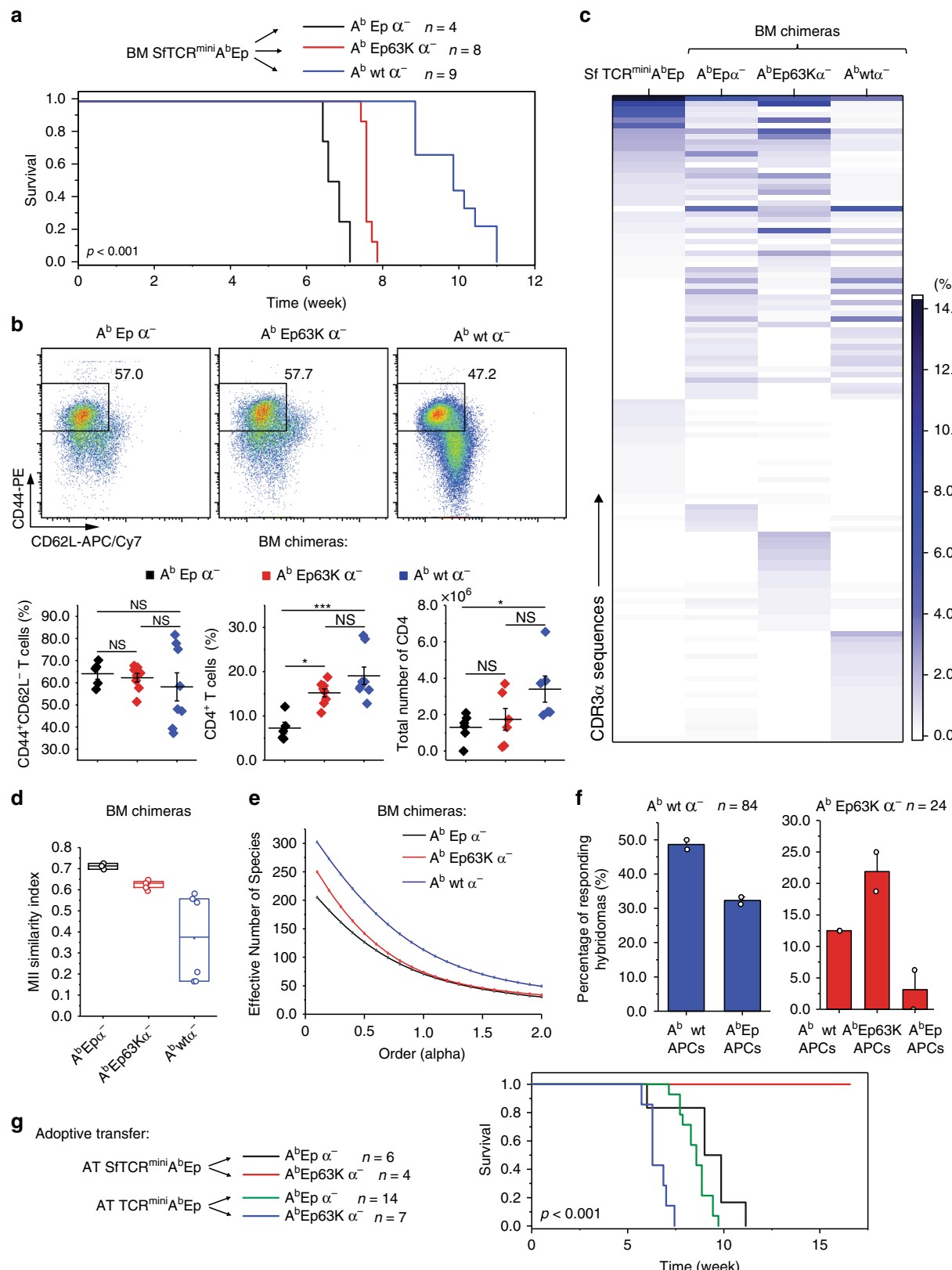

comparison, A$^b$wtTCRα$^-$ recipients manifested the slowest onset and progression of autoimmunity. Lastly, hybridomas established from autoimmune effectors retrieved from A$^b$wtTCRα$^-$ or A$^b$Ep63KTCRα$^-$ recipients responded more frequently to A$^b$wt or A$^b$63K than A$^b$Ep complexes, documenting these TCRs bias toward the positively selecting ligands (Fig. 6f).

In the following experiment, we adoptively transferred CD4$^+$ Foxp3$^{GFP-}$ from SfTCR$^{mini}$A$^b$Ep mice to A$^b$EpTCRα$^-$ or A$^b$Ep63KTCRα$^-$ recipients. In these settings, SfCD4$^+$ cells had been in contact only with A$^b$Ep complexes prior to injection. As shown in Fig. 6g, while the A$^b$EpTCRα$^-$ recipients quickly succumbed to autoimmune disease, the A$^b$Ep63KTCRα$^-$ hosts

**Fig. 6** Thymic and peripheral expression of the same self-peptide accelerates autoimmunity in Sf chimeras. In BM chimeras same donor provided hematopoietic cells expressing only $A^bEp$ complexes whereas radioresistant thymic epithelial cells expressed different ($A^bEp63K$ or $A^bwt$) or the same ($A^bEp$) complexes. **a** Survival curves of $A^bEpTCR\alpha^-$, $A^bEp63KTCR\alpha^-$ and $A^bwtTCR\alpha^-$ mice reconstituted with bone marrow (BM) from SfTCR$^{mini}A^bEp$ mice. **b** Proportions of $CD4^+$ and $CD4^+CD44^+CD62L^-Foxp3^{GFP-}$ cells in different radiation chimeras. **c** Dominant TCRs on $CD4^+$ cells in indicated chimeras (see Supplementary Data 10 for sequence. Unpaired t-test. **d** Similarity indices (MII) for TCR repertoires from SfTCR$^{mini}A^bEp$ mice and various chimeras. **e** Diversity index (REF) for the repertoire of Tregs from different chimeras. **f** Percent of hybridomas established from $CD4^+Foxp3^{GFP-}$ cells from indicated chimeras that responded to $A^bwt$ DCs. **g** Survival curves of $A^bEpTCR\alpha^-$ or $A^bEp63KTCR\alpha^-$ recipients of adoptive transfer (AT) of $CD4^+$ Foxp3$^{GFP-}$ cells from SfTCR$^{mini}A^bEp$ mice. Source data are provided in a Source Data file. Error bars show SD

survived until the end of the experiment (over 16 weeks). These results indicate that the $SfCD4^+$ repertoire recognizes the self-selecting $A^bEp$ complexes as cognate antigens and has impaired ability to sense other $A^b$/peptide ligands. Furthermore, when identical lymphopenic $A^bEpTCR\alpha^-$ or $A^bEp63KTCR\alpha^-$ recipients received an adoptive transfer of naive $CD4^+$ cells from healthy TCR$^{mini}A^bEp$ donors, both types of host mice developed lethal autoimmunity (Fig. 6g). This last result shows that transferred alone $CD4^+Foxp3^{GFP-}$ cells from healthy TCR$^{mini}A^bEp$ mice to lymphopenic recipients expressing exclusively Ep or Ep63K analog caused autoimmunity.

**Identical $CD4^+Foxp3^{GFP-}$ cells from Sf and healthy mice have distinctive gene signatures**. We have shown that autoreactive $CD4^+Foxp3^{GFP-}$ cells from SfTCR$^{mini}$ mice and corresponding cells from TCR$^{mini}$ mice share many identical TCRs that recognize self-peptides as agonists, but the role of these mutual $CD4^+$ clones in autoimmunity and organ damage remain undetermined. Therefore, next we examined the transcriptional signatures of hundreds of individual $SfCD4^+Foxp3^{GFP-}$ cells and their counterparts from control TCR$^{mini}$ mice with a focus on clones expressing the same TCRs to determine if former cells molecular signatures are consistent with pathogenic autoimmune effectors.

For this purpose, we sorted $CD4^+Foxp3^{GFP-}CD44^+CD62L^-$ subsets from SfTCR$^{mini}$ and TCR$^{mini}$ mice and encapsulated over 1500 single cells in microfluidic droplets using a 10× Chromium Controller for parallel scRNAseq analysis and TCR sequencing[32]. Approximately 780 scRNAseq profiles from individual cells of Sf mice and 480 from control mice passed quality control and normalization (see Methods for details). From each of these single cells, we detected an average of 1750 different genes and separately amplified and sequenced their $TCR\alpha$ CDR3 sequences. Next, we performed a t-distributed stochastic neighbor embedding (t-SNE) analysis, which in scRNAseq is often used to separate cell subpopulations according to their distinctive genetic signatures. This analysis showed that $SfCD4^+$ effectors had a plot distribution distinct from most activated but dormant $CD4^+Foxp3^{GFP-}$ cells from control TCR$^{mini}$ mice. Over 80% of individually examined $SfCD4^+$ cells formed two separate clusters that encompassed 307 and 377 individual cells respectively, whereas 70% (361 cells) of corresponding cells from TCR$^{mini}$ mice fell into a third distinct cluster (Fig. 7a). Thus, the three main clusters were dominated by either Sf or control-derived $CD4^+Foxp3^-$ cells, with only a few cross-contaminants (Fig. 7a). The remaining $CD4^+Foxp3^-$ cells (96 cells from SfTCR$^{mini}$ and 83 cells from controls) grouped in a fourth cluster.

Multiple significantly overexpressed genes ($p < 0.001$) could be associated with only one cluster, although the majority of genes were universally represented at comparable levels in all clusters (Fig. 7a–c). For example, the expression of transcription factor *Eomes*, granzymes (k and b), chemokine ligands (*CCL 3, 4,* and *5*), inhibitory and signal regulators (*Ctla2a, Klrg1, Rgs1, SAP, Arap2, TIGIT, Sh2d2a, Fyn, Prkch, Dusp2,* and *CD7*) and integrin *Itga4* were all highly expressed by most cells within the first cluster

(marked in blue) that almost exclusively encompassed half of the Sf-derived effectors (Fig. 7a, b). Overexpression of granzymes, perforin, *CCL3-5*, and *Eomes* is consistent with key functional characteristics of cytotoxic, pathogenic, $CD4^+$ effectors found in multiple sclerosis and systemic fibrotic sclerosis, underlining the autoreactive features of these clones[33,34].

The second cluster (marked in yellow) that encompassed most of the other half of $SfCD4^+$ effectors had discriminatively high expression of interferon-induced *Ifitm1-3, Igtp,* and *Cish*. Less prominently but also significantly overexpressed (from $p < 0.01$ to $p < 0.05$) were *IL7R, Pim1* kinase, *Lgals3, Cxcr6* and *GATA3*, the expression of which in effector $CD4^+$ cells have been linked to altered TCR signaling, inflammation and autoimmune diseases[35,36] (Fig. 7). Finally, the high expression of *Nrp1* and *FR4* (*Izumo1r*) suggested that these cells activation may be impacted by anergy[37].

The majority of activated $CD4^+Foxp3^{GFP-}$ cells from control TCR$^{mini}$ mice formed a separate third cluster (marked in green). Genes discriminatively expressed by this subset include *Stmn1, S100a4, Hmbg2, Vim, Anxa2, Ass1* ($p < 0.001$), which are collectively known to be involved in cytoskeleton reorganization, T cell activation and cell cycle regulation (Fig. 7a, b). The fourth and final cluster (marked in red) grouped activated cells from both strains and was enriched in cells overexpressing *Ly6c* and *CD74* ($p < 0.01$) (Fig. 7).

Differential expression of selected (granzyme B vs Tnfrsf4 and Ly6c vs ICOS) genes was confirmed using flow cytometry, in which staining recapitulated the original split of $SfCD4^+$ effectors into two separate clusters with differential expression of these markers (Supplementary Fig. 13a). Furthermore, overall low expression of Nur77 and moderate levels of CD5 (detected in individual $CD4^+$ and $SfCD4^+$ cells by scRNAseq) correlated with expression of the Nur77$^{GFP}$ reporter and staining by monoclonal antibodies (MoAbs) specific for CD5 (Supplementary Fig. 13b). Although peripheral $SfCD4^+CD44^+CD62L^-$ and $CD4^+CD44^+CD62L^-$ cells express similar levels of Nur77$^{GFP+}$ and CD5$^+$, ex vivo former subset responded to lower concentrations of aCD3 MoAb, suggesting these cells higher sensitivity for weak TCR triggering (Supplementary Fig. 14).

To show that $CD4^+$ cells expressing identical TCRs remain inactive in TCR$^{mini}$ mice, but in SfTCR$^{mini}$ mice convert to autoimmune effectors, we color-coded only these cells on t-SNE plots according to their origin or unique TCR (Fig. 7d, e). Overall, we found 23 TCRs (encoded by 63 different nucleotide sequences) shared by $CD4^+Foxp3^{GFP-}CD44^+CD62L^-$ cells in both mice. These TCRs were expressed by 176 individual cells (123 from SfTCR$^{mini}$ and 53 from TCR$^{mini}$), and more than half of them were also expressed by hybridomas established from $SfCD4^+Foxp3^{GFP-}CD44^+CD62L^-$ cells (Supplementary Data 11). Most $CD4^+$ cells expressing these common TCRs from TCR$^{mini}$ mice co-localized on 2D-visualized t-SNE plots in the third cluster, the transcriptional signature of which was most compatible with an antigen-experienced but dormant phenotype. In contrast, most cells expressing identical TCRs from SfTCR$^{mini}$ mice gathered within either the first or second cluster (Fig. 7d, e).

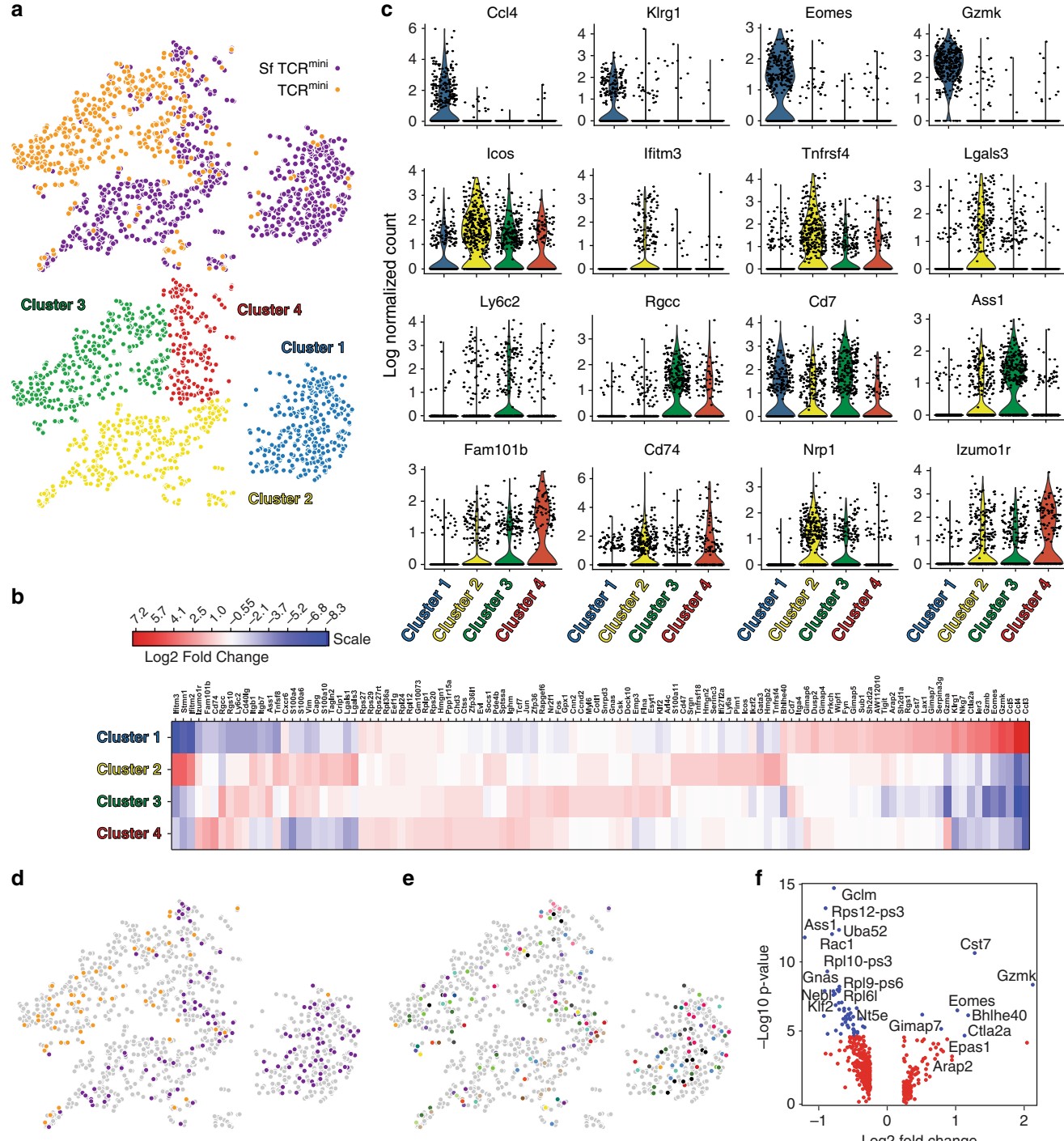

**Fig. 7** Analysis of DE genes in individual SfCD4+Foxp3GFP− and CD4+Foxp3GFP− effectors. **a** 2D visualization of t-SNE plot separating peripheral CD4+ CD44+CD62L−Foxp3GFP− cells from Sf and healthy TCRmini mice. **a** Heatmap of differentially expressed (DE) genes in clustered CD4+Foxp3GFP− cells from SfTCRmini and TCRmini mice. Grid cells are colored by a gene's log2 fold change. **c** Violin plots show the expression of a selected set of DE expressed genes among the different clusters. **d** 2D visualizations of t-SNE projections of individual CD4+Foxp3GFP− cells from TCRmini (orange) and SfCD4+Foxp3GFP− from SfTCRmini (purple) mice that shared 23 common TCRs. **e** 2D imaging of t-SNE dimensionality reduction plot with 23 different TCRs shared by CD4+ single cells in both studied strains. The same color (but not gray) in the plot represents the identical TCR clonotypes. Sequences of TCRα CDR3 regions, coding color and their frequencies in each strain are listed in Supplementary Data 11. **f** Volcano plot shows aggregate of DE genes of all shared clonotypes CD4+Foxp3GFP− cells from SfTCRmini and TCRmini control mice. Genes most relevant to these cells distinctive functional status are specified. Source data are provided in a Source Data file

Further analysis of signature genes using exclusively scRNAseq data from single CD4+CD44+CD62L−Foxp3GFP− cells expressing common TCRs showed that cells from SfTCRmini mice distinctively expressed genes linked to autoimmunity and cytotoxicity (*Gzmk/b*, *Eomes*, *Ctla2a*, *Cst7*). In contrast, the gene signature of corresponding cells expressing the same TCRs from TCRmini mice reflected transcriptional traits of latent CD4+ cells (*Ass1*, *Gclm*, *Rac1*, Fig. 7f). Thus, scRNAseq analysis suggests that

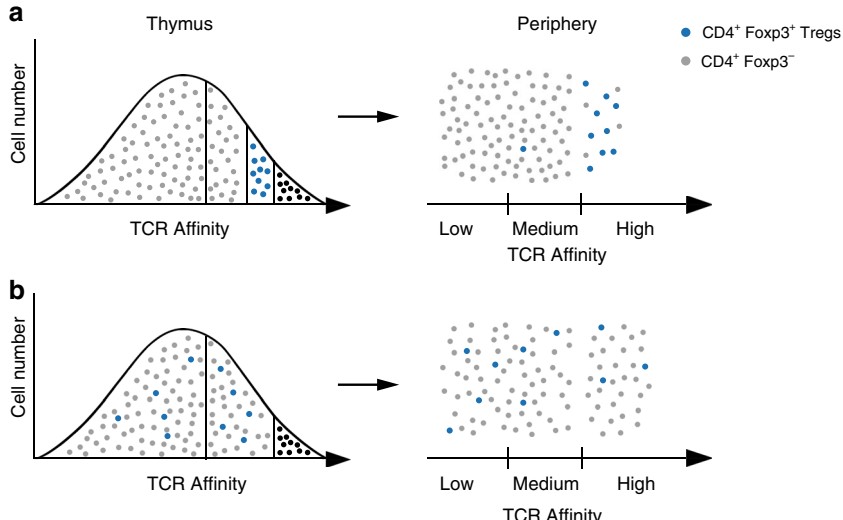

**Fig. 8** Abundance of autoreactive CD4$^+$ T cells-current and proposed hypotheses. **a** Current paradigm: Only thymocytes that TCRs weakly bind class II MHC/ self-peptide complexes differentiate as CD4$^+$Foxp3$^-$ cells (gray dots). Thymocytes that express autoreactive TCRs are deleted by negative selection (black dots) or become CD4$^+$Foxp3$^+$ Tregs (blue dots). ● Negative selection infrequently misses CD4$^+$Foxp3$^-$ cells with autoreactive TCRs. ● In the periphery, Tregs control few (1-4%) autoreactive CD4$^+$Foxp3$^-$ cells. In Tregs deficient mice these rare CD4$^+$Foxp3$^-$ cells clonally expand causing autoimmunity. ● Tregs only complement central tolerance. ● Autoreactivity is a nuance of TCR repertoire of peripheral CD4$^+$Foxp3$^-$ cells but is a common feature of CD4$^+$Foxp3$^+$ repertoire. **b** Proposed model: Approximately 1/3 of mature CD4$^+$Foxp3$^-$ cells express autoreactive TCRs. A considerable portion of CD4$^+$Foxp3$^-$ cells are selected by agonist self-peptides. ● Negative selection fails to delete a substantial portion of thymocytes expressing potentially autoreactive TCRs. ● In Tregs deficient mice autoimmunity involves polyclonal activation of CD4$^+$Foxp3$^-$ cells by ubiquitous self-peptides. ● Central tolerance and Tregs have an equal role in preventing autoreactivity. ● Autoreactive TCRs are expressed at similar frequencies by CD4$^+$Foxp3$^-$ and CD4$^+$Foxp3$^+$cells. TCR self-reactivity has a limited impact on CD4$^+$ cells lineage commitment

in healthy mice many CD4$^+$Foxp3$^-$ appearing as quiescent express autoreactive TCRs that will direct autoimmune responses and self-tissues injury in these cells if peripheral tolerance is compromised.

## Discussion

Comparison of TCR repertoires of Sf and healthy mice strongly suggests that the proportion of potentially autoreactive CD4$^+$ cells is approximately one-fold higher than the highest estimates previously published[15], but not far from estimates of autoreactive CD4$^+$ cells in transgenic mice expressing model autoantigen. In these studies, up to 30% of thymocytes that expressed TCRs specific for model self-peptide survived negative selection and moved to the periphery[7,30]. The final estimate of autoreactive CD4$^+$ cells controlled by peripheral tolerance may exceed 30% of all CD4$^+$ cells as downregulation of inhibitory co-receptors on SfCD4$^+$ cells allows for expansion of autoreactive T cell clones and exacerbated disease symptoms[38–40]. Our data is consistent with the results of the previous study where depletion of Foxp3$^+$ cells in adult mice using DT injections led to accelerated onset of multiorgan, lethal autoimmunity and lymphoproliferation[18]. This report postulated that a very high proportion of CD4$^+$Foxp3 cells is continuously controlled by Tregs. Thus, both unmanipulated C57BL/6 and TCR$^{mini}$ mice consistently show a high prevalence of autoreactive CD4$^+$Foxp3$^-$ cells that were not physically deleted or silenced and need to be controlled by the Tregs.

To show that a substantial proportion of CD4$^+$ cells is not only self-reactive but also pathogenic, we isolated activated effector cells infiltrating various organs from Sf mice and then examined their TCRs and responses to autologous DCs *ex vivo*. Self-reactivity of hybridomas derived from Sf mice (Foxp3$^{GFP-}$ and Foxp3$^{GFP+}$) was somewhat surprising since, in general, hybridomas do not inherit parental CD4$^+$ cell effector features like

Foxp3 expression or Th1/Th2 effector function. This is because intertypic hybrids of equal ploidy (mature CD4$^+$ cell fused with thymoma) fail to express lineage specific traits of either parent[41]. Comparable proportions of hybridomas representing CD4$^+$Foxp3$^-$ cells from Sf and adult DT-treated C57BL/6Foxp3$^{DTR/GFP}$ mice were self-reactive, demonstrating that it is not a unique feature of TCR$^{mini}$ repertoire[22]. However, hybridomas can inherit epigenetically regulated features that influence TCR signaling pathways, suggesting that future studies investigating these cell lines autoresponses should incorporate a detailed analysis of the epigenetic landscape.

Our findings support the previous report that some of the key autoantigens that trigger pathogenesis in Sf mice have ubiquitous expression[42]. We found that around 20% of autoreactive TCRs expressed by dormant CD4$^+$Foxp3$^-$ cells had high functional affinities for these common autoantigens, as reflected by an average fifty-fold increase in Nur77$^{GFP}$ expression and measurable IL-2 secretion by hybridomas established from these cells (Figs. 3 and 5). Relatively high incidence of CD4$^+$Foxp3$^-$ cells that express high(er) affinity, potentially pathogenic TCRs in the peripheral repertoire is therefore significant for our understanding of mechanisms of thymic selection, role of TCR in CD4$^+$ T cell lineage commitment and common features of autoimmune diseases. We propose that the current paradigm of interdependence of central and peripheral tolerance should better emphasize Tregs control over many rather than few autoreactive peripheral CD4$^+$ cells and revisit the role of agonist peptides in positive selection of CD4$^+$Foxp3$^-$ cells (for the proposed model see Fig. 8).

Expression levels of the Nur77$^{GFP}$ reporter, which is proportional to the strength of TCR signaling, were similar in pathogenic SfCD4$^+$ effectors and activated, peripheral CD4$^+$ cells from healthy mice, suggesting that on average, functional affinity of TCRs driving autoimmune responses is not significantly elevated

in Sf mice compared to TCRs expressed by antigen-experienced CD4$^+$Foxp3$^-$ cells in healthy mice. We also found that SfTCR$^{mini}$ mice have few CD4$^+$Foxp3$^-$ cells co-expressing FR4 and CD73, which in C57BL/6 mice mark CD4$^+$ anergic cells, suggesting that Tregs may be also involved in the induction of anergy[37] (manuscript in preparation). In contrast, SfTCR$^{mini}$ mice had twice as many CD4$^+$Foxp3$^-$PD-1$^{hi}$ cells than control mice, suggesting that in some autoreactive clones adversely activated by self-antigens, intrinsic mechanisms of peripheral tolerance attempt to compensate for the lack of functional Tregs. Reportedly, mice with reduced Foxp3 expression in Tregs developed multiorgan autoimmunity when Tregs also lacked PD-1 expression, suggesting that PD-1 inhibition non-redundantly controls an additional pool of autoreactive CD4$^+$ cells[39].

Our results also demonstrate that potentially autoreactive CD4$^+$Foxp3$^-$ cells express abundant TCRs that have been speculated to recognize ubiquitous autoantigens (Fig. 2)[43]. Some of these dominant TCRs were also found in A$^b$-deficient Sf and healthy mice, suggesting that the differentiation and activation of these clones is not strictly dependent on self-antigens presented by class II MHC. These A$^b$ independent clones caused autoimmunity and death of SfTCR$^{mini}$A$^{b-}$ mice after 24 weeks rather than 8–10 weeks as observed in SfTCR$^{mini}$ mice, highlighting the dominant role of self-antigens presented by class II MHC in Sf autoimmunity. Similarly, hybridomas established from SfCD4$^+$Foxp3$^{GFP+}$ cells were also autoreactive, but the significance of these cells in autoimmune diseases remains to be elucidated further[44].

Many studies have concluded that Tregs are particularly important for the induction of tolerance to epitopes from tissue-restricted antigens or neoantigens that are absent or poorly represented in the thymus. Unexpectedly we found that Sf mice lacking a diverse range of self-peptides bound to A$^b$ suffer from accelerated autoimmunity and very severe skin inflammation. The fact that major manifestations in SfTCR$^{mini}$A$^b$Ep mirrored those in SfC57BL/6 mice shows that organ-specific epitopes are not critical to drive autoimmune T cells activation and Sf pathogenesis. The Ep peptide (originally described as Eα(52-68)) is a naturally abundant self-peptide presented by A$^b$ in the thymus and the periphery[45]. We anticipated that SfTCR$^{mini}$A$^b$Ep mice may better tolerate Tregs dysfunction, as central tolerance should have eliminated most of the thymocytes with A$^b$Ep-specific TCRs. On the other hand, in these mice the same self-peptide is not only facilitating both positive and negative selection but is broadly expressed on all class II MHC positive cells in the periphery and can activate self-selected peripheral CD4$^+$ cells. Overall these results suggest a scenario in which lack of functional Tregs leads to polyclonal activation of CD4$^+$Foxp3$^-$ cells that often recognize abundant self-peptides as autoantigens.

## Methods

**Mice.** Animal breeding and experiments were performed in a specific pathogen-free animal facility in compliance with the protocol approved by the Georgia State University Institutional Animal Care and Use Committee, and we complied with all the ethical regulations. Both TCR$^{mini}$, TCR$^{mini}$A$^b$Ep and C57BL/6Foxp3$^{GFP}$ mice were previously described (Pacholczyk 2006, and Jax mice 023800). To introduce *scurfy* mutation in TCR$^{mini}$Foxp3$^{GFP}$ and TCR$^{mini}$A$^b$Ep, mice were crossed with SfC57BL/6 females (Jax 004088) and then intercrossed for 10-12 generations. For adoptive transfer experiments and bone marrow chimeras production TCRα$^-$ (Jax mice 002116), A$^b$EpTCRα$^-$ and A$^b$Ep63KTCRα$^{-31}$ were used. TCR$^{mini}$Nur77$^{GFP}$ mice were obtained by crossing TCR$^{mini}$ with C57BL/6Nur77$^{GFP}$ reporter mice (Jax mice 016617). TCR$^{mini}$Aire$^-$, H2M$^-$ Ii$^-$, and strains were described previously[18,46,47]. To deplete Tregs, C57BL/6Foxp3$^{DTR/GFP}$ mice were injected with diphtheria toxin (50 μg/kg) on five consecutive days[18]. Animals were 6–10 weeks old at the time of experiments (unless otherwise specified) and consisted of males and less often females because matching Sf heterozygote males (Sf mutation in on X chromosome) were used.

**Isolation of T cells from lymphoid and nonlymphoid organs.** Single-cell suspensions were prepared from inguinal and mesenteric lymph nodes by mechanical disruption and passed through 100 μm filter (Corning). Colonic lamina propria T cells were isolated, as previously described[48]. Briefly, colons were opened longitudinally and contents were flushed with ice-cold Hanks balanced salt solution, HBSS (Cellgro). Each colon was cut into small pieces and washed with HBSS solution supplemented with 5% FCS (HyClone) and 2 mM EDTA at 37 °C. A single-cell suspension was obtained after treatment with Collagenase D (1.0 mg/ml) and DNase I (0.1 mg/ml) (both from Roche). A purified and concentrated suspension of lamina propria lymphocytes was obtained after centrifugation on Percoll (GE Healthcare) gradient (45% and 70%). The interface, enriched in leukocytes, was collected and used for experiments. Lungs and liver were harvested, and lymphocytes were isolated by enzymatic digestion for 20 min, using Collagenase D (1.0 mg/ml) and DNase I (0.1 mg/ml) (both from Roche) at 37 °C. For T cell enrichment, Lymphocyte Separation Medium (Corning) was used. The interphase was collected and used for further analysis.

**Flow cytometry and cell sorting.** Monoclonal antibodies conjugated with different fluorescent dyes were purchased from BioLegend, BD or eBioscience unless otherwise listed in the Key Resources Table. Cell surface staining with monoclonal antibodies and intracellular staining for CTLA-4 was done by standard procedures. Samples were analyzed using a CytoFLEX Flow Cytometer (Beckman) or FACS-Canto (Becton Dickinson) and data were processed with FlowJo v10 (FlowJo, LLC). Cells were sorted using Sony SH800 (Sony) and MoFlo cell sorter (Beckman Coulter) with purity above 98%. For the gating strategy see Supplementary Fig. 15.

**Synthesis of cDNA libraries and high throughput sequencing.** Preparation of the library for single-cell was performed from flow-cytometer-purified T cells (purity > 99%), as previously described[48]. Single CD4$^+$Foxp3$^{GFP+}$ and CD4$^+$Foxp3$^{GFP-}$ T cells cells were sorted into 96-well plates from different organs. cDNA was synthesized using MMLV reverse transcriptase (Promega) and random hexamers (Invitrogen) followed by two rounds of PCR via Perfect Taq Polymerase (5 PRIME). Products of CDR3 Vα chain obtained in the second PCR reaction were sequenced. For Ion Torrent high throughput sequencing Cα specific cDNA was synthesized, amplified and CDR3 region of TCRα chain was sequenced[48]. For two-way Illumina HTS library RNA was isolated from the sorted subsets using the RNeasy Mini Kit (Qiagen) according to the manufacturer's procedure. Synthesis of the first complementary DNA (cDNA) strand was performed with a primer specific for the TCR Cα region (5′-TCGGCACATTGATTTGGGAGTC-3′) using Superscript III cDNA synthesis kit (Invitrogen). Incorporation of Illumina sequencing primers together with the bar-coding of DNA material was performed during 2 step PCR amplification using Accuprime Taq Polymerase (Invitrogen). The first step involved amplification and incorporation of Illumina fragment, followed by product purification (NucleoFast 96 PCR (Macherey-Nagel) - removal of primers from previous steps). The PCR reaction was carried out with a pair of primers specific to the Vα2 (5′-ACACTC TTTCCCTACACGACGCTCTTCCGATCTACAGACTCTCAGCCTGGAGACTCA GCT-3′) and Cα (5′-GTGACTGGAGTTCAGACGTGTGCTCTTCCGATCTTTAA CTGGTACACAGCAG-3′) regions of the TCRα chain. During the second PCR step Illumina indexes and adaptors (index 1 with i7 adaptors: 5′-CAAGCAGAAGACG GCATACGAGAT*XXXXXXXX*GTGACTGGAGTTCAGACGTGTGCTCTTCCG ATC-3′ and index 2 with i5 adaptors: 5′-AATGATACGGCGACCACCGAGATCTA CAC*XXXXXXXX*ACACTCTTTCCCTACACGACGCTCTTCCGATCT-3′) where italic font marks an Illumina adaptor sequence. Quality of the library was confirmed by servicing cores using Kapa PCR, Qubit and Fragment Analyzer (High sensitivity NGS fragment analysis) before running it on the Illumina platform. The PCR product was sequenced using the MiSeq Illumina PE250 platform either by Genomics Core at Arizona State University or by Georgia Genomics and Bioinformatics Core at The University of Georgia.

**Bone marrow chimeras.** BM was obtained by flushing donor humerus, tibia, and femur. BM was then RBC lysed and T cell-depleted by labeling cells with biotinylated anti-CD4 and anti-CD8 and anti-biotin microbeads, followed by magnetic cell separation using MS columns (Miltenyi Biotech). In vitro 10 × 10$^6$ cells were injected iv into 750 rad lethally irradiated (RS 2000 Biological Research Irradiator), lymphopenic hosts: A$^b$wtTCRα$^-$, A$^b$EpTCRα$^-$ and A$^b$Ep63KTCRα$^-$ mice. Mice were maintained on antibiotic water (1 mg/ml vancomycin) four days prior and one week after transplantation. Mice weight was monitored every 3 days and euthanized when the bodyweight reached 80% of starting weight.

**Adoptive transfer of Sf CD4$^+$ T cells.** In total 1 × 10$^6$ of naive TCR$^{mini}$Foxp3$^{GFP-}$ CD44$^-$CD62L$^+$ T cells were injected iv to individual 6-week-old, lymphopenic A$^b$EpTCRα$^-$ or A$^b$Ep63KTCRα$^-$ recipients. Bodyweight was monitored every 3 days. Animals that reached a humane endpoint were killed, and no later than after 16 weeks.

**Histology.** Tissues were excised and immediately placed in 10% buffered formalin (Fisher) for at least 48 h to ensure enough penetration, then processed with Excelsior ES tissue Processor (Thermo). Tissues were embedded in paraffin blocks and 4 μm sections were cut with Shandon Finesse 325 Manual Microtome

(Thermo). Slides were stained with hematoxylin and eosin (both from Fisher) according to a standard protocol. Slides were analyzed by an experienced clinical pathologist (N.S.) in a blinded manner. At least three fragments of each tissue were photographed (EVOS FL Auto microscope (×10 magnification) equipped with Pearl Scope software (Fisher). Scale 0-4 was applied: 0-no inflammation (no discernible inflammation), 1-mild (small, focal focus of inflammation), 2-moderate (small, multiple foci of inflammation), 3-strong (multiple large foci of inflammation), and 4- severe (significant inflammation with parenchymal destruction). Data were analyzed with Origin 2017 (OriginLab).

**Hybridoma production and testing.** CD4[+] hybridomas were prepared as reported[49]. The fusion partner was a new variant of BWα[−]β[−] thymoma that has been stably transfected with Nur77[GFP] reporter (BWNur77[GFP], manuscript in preparation). For testing, $2 \times 10^5$ hybridomas were mixed with $2.5 \times 10^4$ BMDCs (6-day old bone marrow cells elicited with GM-CSF (5 ng/ml), >80% CD11c[+]MHCII[+]) in round-bottom wells. As a control, hybridomas were activated with plate-bound aCD3 (10 μg/ml) MoAb or PMA (50 ng/ml) and Ionomycin (0.5μg/ml). After 16 h of Nur77[GFP] upregulation was assessed by FACS in gated CD4[+]Vα2[+] cells. Supernatants were saved for IL−2 detection by HT-2 assay[49]. In selected experiments, hybridomas representing CD4[+]Foxp3[GFP−] cells from DT-treated Foxp3[DTR/GFP] mice or controls were preincubated with GSK503 inhibitor at 5 μM for 24 h before use for activation assay. This preincubation does not change expression of TCR, CD4 or class II MHC.

**Preparation of lysates from non-lymphoid tissues.** Tissues were digested with collagenase D (1.0 mg/ml) and DNaseI (0.1 mg/ml) (both from Roche) for 30 min at 37 °C, washed twice with PBS and resuspended in sonication buffer (50 mM Tris, pH 8.0, 0.05% v/v Tween 20 (all from Fisher), and 1x protease inhibitors (Halt Protease Inhibitor Cocktail, Fisher). Cells were sonicated on ice with Branson sonicator (duty 30, output 2; 1-s burst, 10-s rest, 3 min total). Insoluble material was removed by centrifugation (4 °C, 15 min, $10,000 \times g$). Protein content was measured with the BCA kit (Fisher).

**scRNAseq and TCR sequencing.** Single-cell suspensions were prepared from lymph nodes by mechanical disruption through nylon mesh, washed, counted and stained with appropriate antibodies. CD4[+]CD44[+]CD62L[−]Foxp3[GFP−] and CD4[+]Foxp3[GFP+] populations were obtained by sorting with SH800 sorter (Sony) with purity > 98%. Next cells were subjected to in Gel Bead Emulsion using Chromium 10x Genomics controller according to manufacturer guidelines. To accomplish parallel scRNAseq and TCR sequencing after cDNA amplification, the concentration of each sample was measured using Bioanalyzer (Agilent). Next, up to 50 ng cDNA was used for 5′ gene expression library construction and 1/16 of this amount for TCR V(D)J target enrichment step (using custom-designed primers). To prepare the cDNA libraries for 10x Genomics Chromium controller we used the single-cell 3′ v2 kit. QC for both libraries was done using Bioanalyzer (Agilent) at the GSU Molecular Biology Core and later confirmed by TapeStation 2200 (Agilent) at the University of Florida Interdisciplinary Center for Biotechnology Research. scRNA sequencing was done on Illumina NextSeq500 (1 × 150) machine with a sequencing depth of at least 50,000 reads per cell, whereas TCR library on MiSeq (2 × 150) machine with a sequencing depth of at least 5000 reads per cell. Both were performed at the University of Florida Interdisciplinary Center for Biotechnology Research (Gainesville, FL).

**Processing of scRNAseq data.** Illumina sequencer base call files (BCL) were processed and demultiplexed with CellRanger (v2.2) that used Illumina conversion software bcl2fastq2 (v2.2). FASTQ files were processed using CellRanger software, which performed alignment, filtering, barcode counting, and UMI counting. Single-cell V(D)J sequences were generated with VDJ function of CellRanger using a custom-made VDJ reference genome. For single-cell expression profiling, reads were aligned to the mouse genome (mm10-2.2), with CellRanger counts software, for their assignment as murine sequences. Low-quality cells were excluded in an initial quality control (QC) step by removing genes expressed in fewer than three cells. Also, cells with less than 200 genes expressed or that express >2500 genes were removed. We also removed cells that had >0.05% of mitochondrial-associated genes among their expressed genes. We then normalized the data using log normalization and scale factor 10,000 using the Normalize Data function in Seurat. To identify the differentially expressed features we used the Find Markers function with min.pct argument set to 0.25, which filters out genes expressed in <25% of the cells. PCA, t-SNE and graph clustering were produced using Cell Ranger's pipeline with default settings. Analysis of DE genes downstream was performed using cLoupe (10× Genomics) and R package Seurat software[50].

**Quantification and statistical analysis.** For survival analysis, we used Kaplan-Meier survival curves and p values were calculated using Log Rank Test. For TCR analysis, only CDR3 regions found more than five times in each population per organ and mouse were analyzed. TCRα CDR3 species frequencies were obtained from 2 biological replicates each with 2 technical replicates sequenced both ways. These frequencies were used to determine the number of TCRVα2 CDR3 clonotypes. Statistical significance of the results was performed using Origin Pro (Origin

Lab). The analysis was performed using independent samples *t*-test or a paired sample Student t-test as appropriate. The results were expressed as the mean ± SD unless stated otherwise. *$p \leq 0.05$, **$p < 0.01$, ***$p < 0.001$ as statistically significant, NS-considered as statistically nonsignificant. For the comparative analysis of the overlap, a mutual information index (I-index) was used as defined[25]. Diversity was quantified using the effective number of species (ENS) and presented in the form of diversity profiles (for definitions see[25]).

**Software availability.** Origin 2017 (OriginLab), GraphPad Prism (GraphPad Software) and Adobe Illustrator (Adobe) were used for graphical and statistical analysis (available commercially). The Seurat R package v3.0 was used for scRNA seq and cLoupe (10X Genomics). High throughput sequencing data was processed using custom made Python-based TCR extraction tool and database are available upon request.

**Reporting Summary.** Further information on research design is available in the Nature Research Reporting Summary linked to this article.

## Data availability
scRNAseq sequencing data generated in this study have been deposited in the Gene Expression Omnibus under the primary accession code: = NCBI under the accession number PRJNA575202. The source data underlying Figs. 1a, b, d, e, g, 2a–d, 3a, c, d, f, g, h, 4a, b, d, e, g, 5a, b, 6a–g, 7b and Supplementary Figs. 1–5, 7–13 are provided as a Source Data file. All other data are available on request from the corresponding author. Material Transfer Agreement will be used to transfer any data that can be shared.

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

## Acknowledgements

We thank Ali Foroughi Pour for help with scRNAseq analysis and Dr. Shawna Hiley for editing the manuscript. This work was supported by GSU Institutional funds and NIH grant AI121151 to L.I.

## Author contributions

A.C., M.K., E.S., P.K., and L.I. conceived the study. L.I., A.C., M.K., and E.S. designed experiments, interpreted data and wrote the paper. A.C. performed most presented experiments. N.S. analyzed histological specimens, A.C. and E.S. performed scRNA-seq. M.P., E.S., and G.R. did differential gene expression from scRNAseq. W.E. designed custom software for TCRs analysis and M.P. performed statistical analysis. All the authors read and approved the final manuscript.

## Competing interests

The authors declare no competing interests.

## Additional information

**Supplementary information** is avaliable for this paper at https://doi.org/10.1038/s41467-019-12820-3.

