## [Peer Review File · Nature Communications]

Reviewers' comments:

Reviewer #1, expert in thymic selection (Remarks to the Author):

In this manuscript, the authors make a very convincing argument that a substantial percentage of peripheral CD4+ T cells are in fact pathogenic in the absence of peripheral constraints of Foxp3+ Tregs. It has been long established that in the absence of Foxp3 and functional Tregs, rampant autoimmunity ensues, however it was not known how prevalent dormant pathogenic CD4+ T cells exist in the periphery of healthy mice. The authors make numerous novel and interesting observations. First, through the use of mice that express a restricted TCR repertoire (TCRmini) and breeding onto the Treg deficient, scurfy background, Cebula et al determined that approximately 30% of all peripheral T cells have the potential to cause pathogenic autoimmune responses. Second, the authors sorted CD4Foxp3GFP- T cells and produced CD4+TCR+ hybridomas, some with identical TCR clones that are pathogenic in the absence of Tregs but non-pathogenic in the presence of Tregs. Surprisingly, the hybridomas of T cells derived from scurfy mice (but not the same TCR from healthy mice) kept their high-self reactivity (lower activation threshold), indicating that epigenetic changes may be set during thymocyte development. Last, the authors combined the scurfy-TCR mini mouse with a mouse strain that expresses MHC II (Ab) covalently bound to a single autoantigen (Ealpha). In this design, the authors can examine T cell development against a single self-antigen. Surprisingly, expression of this single antigen caused accelerated autoimmunity in the absence of functional Tregs, indicating thymic central tolerance is not sufficient.

Overall, the manuscript is very insightful however there are a few areas that should be addressed:

Major Comments:

Fig.3: A major finding in this manuscript is that Sf derived CD4 cells have a lower activation threshold to self-antigen than CD4 cells from healthy mice (Section related to Figure 3 and Figure S6). However, there was not a clear mechanism as to how the Sf TCR hybridomas had a lower activation threshold. Related to Figure S6, do SfCD4Foxp3GFP- TCR hybridomas have higher basal (no stim) Nur77 compared to the identical TCR expressed in CD4Foxp3GFP- TCR hybridomas?

Is there an increase in any negative regulators in CD4Foxp3GFP- TCR hybridomas compared to SfCD4Foxp3GFP- TCR hybridomas?

In the SfTCRmini and TCRmini mice, is there a difference in thymic CD4 single positive or peripheral CD5 expression?

This last question may be outside the scope of the manuscript but is there a role for Tregs in maintaining or setting an activation threshold in the thymus?

Fig. 6: The authors state "Next, we investigated the mechanism behind the accelerated onset and progression of autoimmunity.....We hypothesized that when positive and negative selection of thymocytes occurs on the same class II/MHCpeptide complexes, more autoreactive CD4 cell mature...." This is a very important point and should be explored further in the thymus to understand whether the AbEp complexes allow for an increase in positive selection and an increase in more mature CD4SP that leads to "more autoreactive CD4 cells mature" in the periphery. This is related to antigen specific Treg niches. Too much antigen and too little microenvironments for Treg development could overwhelm the Treg/Teff balance as early as the thymus.

Minor Comments:

1) In the results section for Fig. 1A and Fig. 4A, the text of manuscript uses 7-10 weeks and 6 weeks respectively, but the graph is in days, be consistent and pick either weeks or days. Also, the use of dotted lines in Figure 1A for SFWT did not come out on the survival graph, pick a different type of line or use symbols.

2) In the section for Figure 2, which figure or supplemental figure supports the claim from "The shared portion was slightly higher when CD4Foxp3GFP+ repertoires from the strains were compared,..."?

- 3) Clarify which mice were used for the SF mice for Fig. 2A in the text (TCR mini or WT?) in this sentence, "We examined the repertoire of abTCRs expressed by autoreactive CD4 clones in Sf mice, and compared them to"
- 4) For the result "Approximately 20% of identical abTCRs were overrepresented in both sf and healthy mice and could be categorized as public TCRs (found on more than 1% of CD4 cells) (Fig. 2B)". How was the 1% determined. Should star(*) the 20% similar to Fig. 2A
- 5) For Fig. S4C,D,E is there overlap between thymic CD4 single positive TCRs and peripheral TCRs? If there is overlap, why is there no activation of thymocytes.
- 6) Fig. S5 result should be explained further.
- 7) Fig. 4C, for the x-axis, is the Foxp3 staining using intracellular anti-Foxp3 or is this a readout of GFP from Foxp3GFP+. The legend states (F,G) proportions CD4Foxp3GFP+ cells in indicated organs.
- 8) In Fig. 5A, why do only 33% of TCR hybridomas of the AbEp+ T cells respond to AbEp complex when that is the only complex presented, according to the text? Why is it not 100%? Please explain further.
- 9) Fig. 6A, what statistic was used for the p value and which groups are being compared?
- 10) Fig. 7B, please explain the comparison for the heat map. How was the fold change calculated, against which group?
- 11) Check grammar (use of commas) and spelling throughout. It is not possible to point out the issues without line numbers. Example is on the first page of Results section "sf mic" and second page of results, after (Fig. S4A) change the has to was.

Reviewer #2, expert in sc-RNA seq and T cell repertoire (Remarks to the Author):

The manuscript "Dormant pathogenic CD4 T cells are prevalent in the peripheral repertoire of healthy mice" by Cebula et al. provides insights into the development and regulation of the self-reactive CD4 T cell repertoire. Using a genetic mouse model with a restricted T cell repertoire, high-throughput sequencing and in vitro experiments the authors suggest that more than a third of the CD4 T cell repertoire in mice is auto-reactive and potentially pathogenic. They authors suggest that such dormant repertoire is constantly repressed by regulatory T cells that limit the activity of pathogenic CD4 T cells recognizing self antigens.

Specific comments:

1/The authors propose that more than 30% of the CD4 T cells are self-reactive in a normal repertoire by comparing CD44+ cells from Scurfy mice to non Treg from WT mice. Although suggested by the results derived from the TCR repertoire in the Scurfy mice (with limited TCR repertoire) none of the hybridoma derived from WT mice seems to respond to autologous APC. Does this suggest that the Scurfy repertoire massively overestimate this number in WT mice? How this result fits into the author's conclusion is unclear and should be address experimentally.

2/In the TCR-restricted setting, the authors show that while hybridomas derived from Scurfy animals respond to self-antigens, the same hybridomas (1-5 Figure 3), expressing the same TCR sequences failed to do so. Although interesting, these results are puzzling in the absence of potential mechanism explaining the differential response of identical TCR. Would it be possible that these hybridomas express alternate TCR α chains that will confer differential responsiveness to self peptide ? Alternatively, is the level of TCR expression and peptide-MHC interaction similar ? Are the hybridoma from Scurfy present different gene expression or chromatin accessibility than the ones from WT ?

3/The volcano plot in Fig7F shows differentials expressed genes of the shared TCR clone type. Are these an aggregate of all shared clonotypes in the Scurfy vs WT mice ? Is this showing that Scurfy cells are more activated than they WT counterparts (which is expected) ?

4/A number of references are inaccurate and/or missing. Examples Satija 2015, Zemmour 2018, Singh 2011, Cebula 2013 ...

5/ Errors in the text:

"This is because inflammation in Sf mic, releases"

"and BC57BL/6 mice activate 30% of peripheral SfCD4 cells."

"[...] used for 5' gene expression library construction and 1/16 for TCR V(D)J target enrichment step (where custom-designed primers were used). We used the 10x Genomics Chromium single-cell 3' v2 kit ...".

"Cell Ranger's pipeline with default settings using coupe (10X Genomics)"

Reviewer #3, expert in mechanisms of tolerance (Remarks to the Author):

Comments to authors:

This is an interesting study which aims to determine the number of autoreactive CD4 T cells present in a mature repertoire that under normal circumstances are prevented from becoming overtly autoreactive, as well as their functional and CDR3 sequence characteristics when quiescent as compared to autoimmune mice which lack regulatory T cells. While the experimental tools being used are creative, such as the ability to perform sc-RNAseq analysis of cells where the CDR3 can also be determined in mice with and without regulatory T cells, the quantitative arguments being made are not compelling, in part as a result of a lack of sufficient detailed explanation of the analyses performed and in part due to the way the data is presented and summarized, yet this is clearly where the novelty of this study lies. Fuzzy use of terminology, vague arguments, and very incomplete explanations do not do the data generated in this study justice and make the conclusions drawn very hard to follow. Moreover, it is not always clear why particular experiments are being performed with regard to the big picture. The manuscript would benefit greatly from revisions that improve its clarity and precision, as well as better highlighting the narrative thread from one figure to the next.

Major comments:

Introduction:

- Can the authors comment on where the estimate of 4% autoreactive CD4 T cell clones comes from – ie, how this was measured? To what extent is this estimate exaggerated by the fact that inflammation/tissue damage gives T cells access to additional self antigens that they would not normally be able to gain access to? (Of note, the authors briefly refer to this caveat later in the results section #2).
- It would be helpful to define exactly what the authors mean by "dormant clones". Are they limiting this definition to cells held in check by Tregs or using this more broadly as any autoreactive cells not active for a variety of reasons (including cells not currently exposed to possible self antigens)? Having defined it precisely, this becomes a more useful term throughout the manuscript and can be used to explain exactly how the number of dormant clones is being calculated (and clarifies what caveats the calculations made might have – reasons for over/under-estimations).
- Could the authors describe the TCR-mini mice more fully with regard to the repertoire diversity reduction relative to normal mice? Eg. paragraph 1 of the discussion should be in the introduction instead.

Figure 1

- The claim that in Nur77-gfp mice "autoimmune effectors can be separated according to TCR signal

strength" needs to be substantiated by data or references. This was not done in the Nur77 papers cited (Moran 2011, Kuczma 2009). Specifically, how can Nur77 be used to distinguish autoreactive from other effector cells, as is being implied here?

- Given that in all text references to time points the authors use weeks, it would be helpful if the survival graph in Fig 1A (and Fig 4A) is also shown in weeks. Notably, it is impossible to distinguish the lines for sfTCR-mini and sfWT (they both look identically black).
- Why is there a difference in onset of autoimmunity between sfTCR-mini and sfWT mice? This is not commented on, but the shift is substantial. How does this relate to later arguments (Fig 6 onwards) made with regard to peptide-MHC that T cells are being positively and negatively selected on, if at all?
- What is the rationale for showing the increased percent as well as total numbers of CD4 T cells in tissue in sfTCR-mini mice (Fig 1e)? The authors do not explain this. It seems that the latter (cell counts) is enough to make the point (and it would be clearer if this was shown on a log-scale) of greater T cell infiltration of tissues in conjunction with the activation data from Fig 1c,d.
- The authors state that "the proportion of CD4 cells expressing FoxP3-GFP was lower than in healthy mice" in sfTCR-mini mice, but the data shown in Fig 1F,G only substantiates this in the colon. Could the authors please clarify?

Figure 2

- This figure is not easy to follow due to a lack of details regarding the experiment performed and the data being shown. Specifics to clarify:
 - o Are the authors assessing the TCRalpha and beta chains on a single-cell level from individual cells? I.e, is the "CDR3 sequence" shown in B the joint alpha-beta pair? If so, how are CDR3 being considered where there are >1 alpha chain in the cell?
 - o The stars in Fig 2A are not explained until the next results section, making this confusing. The authors should consider presenting this aspect in Fig 3 instead.
 - o How many mice from each strain were sequenced? In the calculation of overlap between strains, how does this compare to overlap between mice? (also for MI index, this is important to understand). What is the increase in overlap as a result of outgrowth or autoreactive clonotypes, rather than just overlap due to reduced repertoire diversity?
 - o Given that the goal of this analysis is the comparison of sequence presence/absence and frequency between strains, it would perhaps be more useful to plot the concordance (ie, TCRmini on one axis and sfTCRmini on other axis with heat map to indicate the correlation rather than overall frequency). The heatmaps as they stand are difficult to interpret with regard to quantitative patterns observed and the main text does not provide enough explanation.
 - o In the statement "we examined the repertoire of abTCRs expressed by autoreactive CD4 T cell clones in Sf mice", what evidence is there that these are "autoreactive"? Is this an inference based on increase in frequency? If so, this should be clarified and explained why this is a valid assumption.
 - o Fig 2B is cited with reference to a statement about "public TCRs" but this is not data that is included in the figure.
- The conclusion that "one third of CD4FoxP3- cells from healthy TCRmini expressed potentially autoreactive abTCRs" is not substantiated by the data. It is not clear where this number comes from, and whether the caveats listed above have been considered in this analysis.

Figure 3

- In this sentence "analyzed hybridomas have not made IL-2 constitutively and expressed only.." (grammar issues), could the authors clarify whether they mean that in the absence of stimulation the hybridomas do not respond either by IL2 production or by Nur77 induction?
- It would be good to include some raw data examples from the hybridoma experiment that is summarized in A, both with regard to the IL2 response and the Nur77 upregulation.
- What was considered background with regard to scoring hybridomas as responsive or unresponsive to autologous APC stimulation? Were the BMDCs/splenocytes activated?
- Why is the same TCR sequence hybridoma from an SfTCR-mini mouse responding while from a TCR-mini mouse it is not? Is this a result of inhibitory receptors expressed by the cell? Does this not imply

that these T cells are fundamentally different and it is not simply the absence of Tregs which releases the cells from "dormancy"?

Figure 4

- It is unclear why the authors do the experiments presented here. Why would the data shown in Figs 1-3 naturally lead to experiments designed to restrict the TCR repertoire to recognition of a single peptide? Can the authors make more explicit what question is being answered here? Currently it seems more like these mice were in-hand and thus examined, rather than there being a particular hypothesis being tested with regard to the presence of dormant CD4 T cells in the repertoire.
- Why would the expression of a single autoantigen "improve the stringency of negative selection" rather than greatly curtail positive selection? In fact, doesn't the accelerated autoimmunity in these mice suggest that negative selection is less stringent?
- What is "rambling autoimmunity"?
- Why is the autoimmunity in sfTCR-miniAbEp mice unexpected, given previous work showing that the repertoire in single-peptide mice generates T cells which are very MHC-reactive? (Lucas and Germain, 1996 for a review).
- Also see Fig1 comments with regard to some of the data shown here that applies to this figure as well.

Figure 5/6

- The hypothesis presented (that more autoreactive CD4 cells mature and require continual suppression by Tregs if selected positively and negatively on the same peptide-MHC complex) would be more compelling if the data clearly showed that the characterized autoreactive cells are being suppressed by T cells, rather than pushed into autoreactivity by the inflammatory milieu and tissue damage in Treg-deficient mice. The authors should also explain why this particular hypothesis is proposed. What are possible alternatives?
- Labeling in the figure to make clear which bone marrow chimeras result in which outcome with regard to positive and negative selection on what cells would greatly aid in the clarity of figure 6.

Figure 7

- Clusters 1-4 should be labeled in Fig 7A
- By limiting the comparison here to activated T cells in the TCR-mini strain, are the authors not missing the majority of truly 'dormant' (still naïve) cells? Would the majority of the gene signature differences in this comparison not simply be a result of the comparison of true effector cells actively responding to peptide-MHC with quiescent, memory-phenotype cells generated for instance by neonatal lymphopenia-induced proliferation?
- What is the y-axis in Fig 7C? fold change? CPM?
- Fig 7F, which way is the data plotted with regard to fold change (right side are sfTCRmini or TCRmini upregulated genes)?
- How many genes are different between cells with shared TCRs from the two strains? The conclusion that cells are autoreactive in the sfTCR-mini compared to the TCRmini based on the extensive data collection at the single-cell level is underwhelming. What did the sc-RNA level analyses reveal beyond that the cells in sfTCR are actively responding while those in the TCRmini strain are not? These analyses seem primarily confirmatory rather than revealing anything the authors did not already know.

Discussion: there is an entire paragraph on PD1 expression but no data on this was included in the main manuscript. Thus none of the unpublished results cited here can be evaluated. The authors should focus their discussion on data that is shown in the manuscript.

A summary model schematic would be very helpful.

Minor comments:

There are quite a few grammar issues/typos that should be corrected (some examples listed here from the start of the manuscript, but by far not exhaustive as there are too many):

- Intro: para 1, Line 8 "although for these cells in vivo pathogenicity has not been examined"
- Intro: para 1, last line "but that global identification.."
- Intro: para2, line 1: grammar "and the presence of tregs which continuously suppress these cells."
- Intro: para 2, line 7: Fatal "outbreaks" of autoimmunity? This is surely the wrong word.
- Intro: last para "transpire to the peripheral repertoire". What is meant here?
- Intro: last para, final sentence "dynamically modulate the activation threshold"
- Results section #2, first paragraph, line 2: "This is because inflammation" typo in mice.

**Center for Translational Immunology
Institute for Biomedical Sciences**

*Leszek Ignatowicz, PhD
Professor*

July 29, 2019

NCOMMS-19-14032

Dear Editor and Referees,

Please find enclosed our revised paper entitled "Dormant pathogenic CD4 T cells are prevalent in the peripheral repertoire of healthy mice". To address Reviewers concerns, we carefully amended our manuscript and added two new results. First, we demonstrate that one-third of hybridomas representing CD4Foxp3⁻ cells from toxin treated adult C57BL/6Foxp3^{DTR/GFP} mice are activated by autologous DCs (see Fig. 3G,H). This result shows that CD4Foxp3⁻ subset that expresses WT TCR repertoire contains a comparably high proportion of dormant, autoreactive clones as originally found in TCR^{mini} mice that express limited TCR repertoire. Second, we report that activation of autoreactive hybridomas can be attenuated by a pharmacological inhibitor of EZH2 lysine methyltransferase that regulates cytosolic polycomb repressive complex 2 (PRC2) (see new Fig. S11). Second result suggests that dissimilar responses of hybridomas derived from Tregs-sufficient vs Tregs-deficient mice to C57BL/6 DCs reflect differences in epigenetic regulation of TCR signaling in parental CD4Foxp3⁻ cells.

We would like to thank all Referees for their comments how to improve this report. Below please find our point-by-point response to Referee's comments. All changes in the text are underlined.

Sincerely,

Leszek Ignatowicz, PhD

Reviewer #1:

Major Comments:

Fig.3: A major finding in this manuscript is that Sf derived CD4 cells have a lower activation threshold to self-antigen than CD4 cells from healthy mice (Section related to Fig. 3 and Fig. S6). However, there was not a clear mechanism as to how the Sf TCR hybridomas had a lower activation threshold.

We found that lower activation threshold of CD4Foxp3^{GFP-} cells hybridomas from Sf mice as compared to corresponding hybridomas from healthy mice (with wild type or TCR^{mini} repertoire) was not due to different expression of TCR,CD4,PD-1 or CTLA-4. Activation by MoAb aCD3 measured by change in Nur77^{GFP} expression and IL-2 production was also similar, and basal levels of Nur77^{GFP} reporter in compared hybridomas were similar (see new Fig. S4, 5, 6 and 7).

However, in the revised manuscript we show that autoreactive responses of CD4Foxp3^{GFP-} hybridomas were reduced by specific inhibitor of Ezh2 (new Fig. S11). Reportedly, cytosolic Ezh2-associated methyltransferase protein complex controls ligand-induced intracellular signaling in T cells¹. The same inhibitor (GSK503) also reversed autoimmunity in B6Foxp3^{DTR/GFP} mice after Tregs depletion². Thus, the differences in activation of CD4

Georgia State University
Research Science Center, Room 314
100 Piedmont Avenue
Atlanta, Georgia 30303

Tel: 404-413-6685
Fax: 404-413-3580
E-mail: lignatowicz@gsu.edu

hybridomas from *Sf*(or $\text{Foxp3}^{\text{DTR/GFP}}$) and control mice likely reflect changes in epigenetic regulation of TCR-mediated signaling.

Related to Fig. S6, do *Sf*CD4Foxp3GFP- TCR hybridomas have higher basal (no stim) Nur77 compared to the identical TCR expressed in CD4Foxp3GFP- TCR hybridomas? Overall, expression of Nur77^{GFP} by unstimulated hybridomas from *Sf*CD4Foxp3^{GFP-} or CD4Foxp3^{GFP-} cells from AireTCR^{mini} mice was comparable, and parental cells from these strains had overlapping expression of this reporter. However, individual hybridomas had differences in basal Nur77^{GFP} expression that was not correlated with their origin. Therefore, a response of each hybridoma is presented as a fold change over the background (basal expression of Nur77^{GFP} by unstimulated hybridoma). Examples of Nur77^{GFP} expression from *Sf*CD4Foxp3^{GFP-} hybridomas are presented on new Fig. S4 and S6.

Is there an increase in any negative regulators in CD4Foxp3GFP- TCR hybridomas compared to *Sf*CD4Foxp3GFP- TCR hybridomas? CD4Foxp3GFP-. We found no correlation between activation of hybridomas from *Sf* or toxin treated $\text{Foxp3}^{\text{DTR/GFP}}$ mice to C57BL/6 DCs and reduced expression of PD-1 or CTLA-4 (see new Fig. S7).

In the *Sf*TCRmini and TCRmini mice, is there a difference in thymic CD4 single positive or peripheral CD5 expression? We found no difference in CD5 expression on compared peripheral CD4Foxp3⁻ cells, but TCR^{mini} thymocytes had slightly higher mean value for this marker (see below and Fig. S13B).

This last question may be outside the scope of the manuscript but is there a role for Tregs in maintaining or setting an activation threshold in the thymus? We are not aware of published data that has directly examined this issue.

Fig. 6: The authors state “Next, we investigated the mechanism behind the accelerated onset and progression of autoimmunity.....We hypothesized that when positive and negative selection of thymocytes occurs on the same class II/MHC peptide complexes, more autoreactive CD4 cell mature....” This is a very important point and should be explored further in the thymus to understand whether the AbEp complexes allow for an increase in positive selection and an increase in more mature CD4SP that leads to “more autoreactive CD4 cells mature” in the periphery. This is related to antigen specific Treg niches. Too much antigen and too little microenvironments for Treg development could overwhelm the Treg/Teff balance as early as the thymus. A^bEp mice that express unmanipulated or TCR^{mini} repertoires remain healthy over their lifespan and do not develop autoimmunity, suggesting that their thymic microenvironment produces a balanced number of autoreactive and Tregs^{3,4}. In our opinion, data from *Sf*A^bEp mice shows that abrogating Tregs can incite an autoimmune response of CD4Foxp3⁻ cells to originally positively selecting self-peptide. We hypothesize that approximately one third of CD4Foxp3⁻ cells express TCRs selected on agonist self-peptides, but these autoreactive cells remain concealed by Tregs. How often autoreactive CD4Foxp3⁻ cells from *Sf*C57BL/6 mice are activated by originally selecting peptide vs other self-peptides, remains an open question.

Minor Comments:

1) In the results section for Fig. 1A and Fig. 4A, the text of manuscript uses 7-10 weeks and 6 weeks respectively, but the graph is in days, be consistent and pick either weeks or days. Also, the use of dotted lines in Fig. 1A for SFWT did not come out on the survival graph, pick a different type of line or use symbols. Following Referee suggestions, we changed the description for X-axis on all survival graphs (to weeks) and used different graphics to depict survival curves.

2) In the section for Figure 2, which figure, or supplemental figure supports the claim “The shared portion was slightly higher when CD4Foxp3GFP⁺ repertoires from the strains were compared,...”? We apologize but this should read “when CD4Foxp3^{GFP-} repertoires from the strains were compared”. As also suggested by Ref.3 we now provide plots showing 50 dominant CD4Foxp3⁻ clones in different organs in both mice and amended MII similarity indices (see new Fig. S2 and Fig.2B).

3) Clarify which mice were used for the SF mice for Fig. 2A in the text (TCRmini or WT?) in this sentence, “We examined the repertoire of abTCRs expressed by autoreactive CD4 clones in Sf mice and compared them to” Fig.2A shows the allocation of 50 most dominant TCRs from SfTCR^{mini} on respective subset in TCR^{mini} mice because restricted TCR repertoire allows tracking cells expressing the same TCRs in different organs and CD4 subsets. Such comprehensive analysis will be very difficult to accomplish in SfC57BL/6 and control mice expressing WT repertoire.

4) For the result “Approximately 20% of identical abTCRs were overrepresented in both sf and healthy mice and could be categorized as public TCRs (found on more than 1% for CD4 cells) (Fig. 2B)”. How was the 1% determined. Should star(*) the 20% similar to Fig. 2A. To improve clarity, we moved former Fig.2A to Fig.3 to emphasize that identical TCRs were also found on autoreactive hybridomas. We also replace term “public” with term “dominant” or “abundant” to describe most common TCRs. Percent of given TCR is calculated by dividing counts of TCR by total number of TCRs in given repertoire multiplied by 100. Asterisk (*) marks TCRs also found expressed on autoreactive hybridomas produced from SfCD4Foxp3⁻ cells.

5) For Fig. S4C,D,E is there overlap between thymic CD4 single positive TCRs and peripheral TCRs? If there is overlap, why is there no activation of thymocytes. TCRs expressed by CD4Foxp3^{GFP-} thymocytes and peripheral cells from TCR^{mini} mice overlap significantly⁵, but we have not sequenced repertoire expressed by SfTCR^{mini} thymocytes. We have examined over 150 hybridomas made from SP SfCD4Foxp3^{GFP-} thymocytes that reproducibly responded to aCD3 MoAb activation but not to C57BL/6 DCs. Possibly activation threshold of SfCD4Foxp3⁻ cells for autoantigens changes following these cells output to the periphery.

6) Fig. S5 result should be explained further. Explain in text that SfTregs area also self-reactive (results or discussion?). We amended the discussion and confirmed that disabled SfCD4Foxp3^{GFP+} cells are also self-reactive as detected by Tregs-derived hybridomas.

7) Fig. 4C, for the x-axis, is the Foxp3 staining using intracellular anti-Foxp3 or is this a readout of GFP from Foxp3^{GFP+}. The legend states (F,G) proportions CD4Foxp3^{GFP+} cells in indicated organs. Corrected legend and x-axis description now consistently read that we studied a readout of GFP from Foxp3^{GFP} reporter.

8) In Fig. 5A, why do only 33% of TCR hybridomas of the AbEp⁺ T cells respond to AbEp complex when that is the only complex presented, according to the text? Why is it not 100%? Please explain further. We hypothesize that A^bEp selects thymocytes which TCRs that bind this complex with a wide affinity range (from low to high). Most thymocytes with high(er) affinities TCRs die upon negative selection but around 30% escape deletion and in

SfA^bEp mice these escapees respond to *A^bEp* complexes (Fig. 5B). We anticipate that remaining quiescent CD4Foxp3⁻ cells express low(er) affinity TCRs or stay controlled by other mechanisms of peripheral tolerance.

9) Fig. 6A, what statistic was used for the p value and which groups are being compared? Student test?

For survival analysis, we used Kaplan-Meier survival curves and p values were calculated using Log Rank test. This information was also added to the Methods section.

10) Fig. 7B, please explain the comparison for the heat map. How was the fold change calculated, against which group? The heatmap displays 100 most significant (up and downregulated) genes per cluster. Each column shows expression of a significant feature, and each row represent a cluster. Grid cells are colored by a gene's log₂ fold change in its cluster row, compared to the other clusters.

11) Check grammar (use of commas) and spelling throughout. It is not possible to point out the issues without line numbers. Example is on the first page of Results section “*sf mic*” and second page of results, after (Fig. S4A) change the *has* to *was*. We have revised the grammar and examined text to eliminate errors.

Reviewer #2,

1/The authors propose that more than 30% of the CD4 T cells are self-reactive in a normal repertoire by comparing CD44⁺ cells from Scurfy mice to non Treg from WT mice. Although suggested by the results derived from the TCR repertoire in the Scurfy mice (with limited TCR repertoire) none of the hybridoma derived from WT mice seems to respond to autologous APC. Does this suggest that the Scurfy repertoire massively overestimate this number in WT mice? How this result fits into the author's conclusion is unclear and should be address experimentally.

In general, hybridomas representing CD4Foxp3^{GFP-} T cells from WT mice do not secrete IL-2 in response to autologous APCs , because parent CD4⁺ T cells are tolerant to self-peptides and this unresponsiveness is transmitted to hybridomas. Thus, CD4 hybridomas from WT mice are routinely used to detect response to foreign peptides presented by autologous APCs with no background response to APCs only. Rare CD4 hybridomas from WT mice that made IL-2 in response to autologous APCs predominantly recognized FCS-derived peptides from the culture medium (Pullen and Munro, 1988), which in the past was interpreted as autoreactive responses. No hybridomas from WT mice made IL-2 when incubated with autologous DCs, although few had increased expression of Nur77^{GFP} as seen in T cells from Nur77^{GFP} reporter mice⁶.

2/In the TCR-restricted setting, the authors show that while hybridomas derived from Scurfy animals respond to self-antigens, the same hybridomas (1-5 Figure 3), expressing the same TCR sequences failed to do so. Although interesting, these results are puzzling in the absence of potential mechanism explaining the differential response of identical TCR. Would it be possible that these hybridomas express alternate TCRα chains that will confer differential responsiveness to self-peptide ? Alternatively, is the level of TCR expression and peptide-MHC interaction similar? Are the hybridoma from Scurfy present different gene expression or chromatin accessibility than the ones from WT? Second TCRα chains were very rare (less than 1%) as detected by single-cell TCR sequencing. Expression of more than one TCRα chain per cell disturbs read of CDR3 region upon single cell TCRα sequencing, and other TCRβ chains were also rare, (⁵supplemental material). Levels of TCRs and co-receptors were very similar on all compared hybridomas (Fig.S4). We continue to examine differential gene expression and chromatin accessibility in these cell lines to decipher relevant mechanisms. In the revised manuscript we also show that lack of Tregs impacts TCR signaling via polycomb repressive complex 2 (PRC2)², because pre-treatment with EZH2 inhibitor interferes with autoreactive activation of CD4Foxp3⁻ hybridomas by C57BL/6 DCs (see new Fig. S11). We also show that *SfCD4Foxp3⁻* cells have an increased sensitivity to stimulation by low concentrations of aCD3 MoAb, suggesting that these cells activation threshold is lower (Fig. S14).

3/The volcano plot in Fig7F shows differentials expressed genes of the shared TCR clone type. Are these an aggregate of all shared clonotypes in the Scurfy vs WT mice ? Is this showing that Scurfy cells are more activated than they WT counterparts (which is expected)?

The Reviewer is right that volcano plots show DE genes as aggregates for all shared clonotypes from the Sf and WT mice, and that the former cells are activated effectors. We included this information in the revised legend in Fig. 7.

4/A number of references are inaccurate and/or missing. Examples Satija 2015, Zemmour 2018, Singh 2011, Cebula 2013 ...All references were edited and if necessary corrected.

5/ Errors in the text:

“This is because inflammation in Sf mic, releases”

“and BC57BL/6 mice activate 30% of peripheral SfCD4 cells.”

“[...] used for 5' gene expression library construction and 1/16 for TCR V(D)J target enrichment step (where custom-designed primers were used). We used the 10x Genomics Chromium single-cell 3' v2 kit ...”.

“Cell Ranger’s pipeline with default settings using cLoupe (10X Genomics)”

We apologize for errors and these typos were corrected.

Reviewer #3,

The manuscript would benefit greatly from revisions that improve its clarity and precision, as well as better highlighting the narrative thread from one figure to the next. We have edited our report to improve its clarity and precision, and to incorporate all Referee’s suggestions.

Major comments:

Introduction:

- Can the authors comment on where the estimate of 4% autoreactive CD4 T cell clones comes from – i.e., how this was measured? To what extent is this estimate exaggerated by the fact that inflammation and tissue damage gives T cells access to additional self-antigens that they would not normally be able to gain access to? (Of note, the authors briefly refer to this caveat later in the results section #2).

To estimate autoreactive CD4 T cells previous studies followed an increase in CD69 or Nur77^{GFP} expression, CSFE dilution, or precursor frequencies ^{7,8}. Others used staining with class II MHC/self-peptide tetramers ⁹ or followed the allocation of public TCRs ¹⁰. For testing self-reactivity, we measured hybridoma responses to BM-derived or splenic DCs from healthy C57BL/6 that present the physiological spectrum of self-peptides.

- It would be helpful to define exactly what the authors mean by “dormant clones”. Are they limiting this definition to cells held in check by Tregs or using this more broadly as any autoreactive cells not active for a variety of reasons (including cells not currently exposed to possible self-antigens)? Having defined it precisely, this becomes a more useful term throughout the manuscript and can be used to explain exactly how the number of dormant clones is being calculated (and clarifies what caveats the calculations made might have – reasons for over/under-estimations). As dormant CD4Foxp3⁻ cells, we referred to cells which tolerance to self-peptides depends on functional Tregs. Other mechanisms of peripheral tolerance that may depend on Tregs (anergy?) can also play a role. This statement is provided in the Introduction (second paragraph, last sentence).

Frequencies of autoreactive CD4Foxp3⁻ cells in healthy TCR^{mini} (or TCR^{mini}A^bEp) mice were estimated by calculating the (%) of CD4Foxp3⁻ cells that expressed the same TCRs as SfCD4Foxp3⁻ hybridomas activated by C57BL/6 DCs. First, we identified autoreactive hybridomas, next sequenced their TCRs and finally extrapolated these sequences to a database of TCRα CDR3 regions obtained by HT or single cells sequencing.

For *Sf*C57BL/6 and C57BL/6Foxp3^{DTR/GFP} mice that expressed WT repertoires estimate of autoreactive cells reflects the percent of hybridomas established from pLNs CD4Foxp3⁻ cells activated by C57BL/6 DCs. Please note that very similar percent of CD4Foxp3⁻ autoreactive hybridomas were found in mice expressing restricted (*Sf*TCR^{mini}, *Sf*TCR^{mini}A^bEp) or WT (*Sf*C57BL/6 and DT/C57BL/6Foxp3^{DTR/GFP}) repertoires, demonstrating that it is not a unique feature of TCR^{mini} strains. We also tested hybridomas responses to two tissue-specific lysates and *Sf*-derived DCs, but these assays did not change original estimates.

- Could the authors describe the TCR-mini mice more fully with regard to the repertoire diversity reduction relative to normal mice? E.g. paragraph 1 of the discussion should be in the introduction instead. WT mice are expected to harbor around 10⁷ of different TCRs¹¹, whereas from TCR^{mini} mice we have sequenced approximately 10⁵ of different TCRs (CDR3 α). Revised opening paragraph in the Results section provides a detailed description of TCR^{mini} mice and original references.

Fig. 1

- The claim that in Nur77-gfp mice “autoimmune effectors can be separated according to TCR signal strength” needs to be substantiated by data or references. This was not done in the Nur77 papers cited (Moran 2011, Kuczma 2009). Specifically, how can Nur77 be used to distinguish autoreactive from other effector cells, as is being implied here? We apologize for the misunderstanding. We meant to separate autoreactive CD4⁺ T cells expressing high vs low level of Nur77^{GFP} reporter, rather than to separate self-reactive clones from other CD4 cells. Expression profiles of Nur77^{GFP} reporter by activated CD4 cells from *Sf* and healthy controls are mostly overlapping (Fig. S13B).

- Given that in all text references to time points the authors use weeks, it would be helpful if the survival graph in Fig 1A (and Fig 4A) is also shown in weeks. Notably, it is impossible to distinguish the lines for *Sf*TCR-mini and *Sf*WT (they both look identically black). We changed survival curves graphics and duration to weeks. We also modified depicting lines, as collectively suggested by Referee 1 and 3.

- Why is there a difference in onset of autoimmunity between *Sf*TCR-mini and *Sf*WT mice? This is not commented on, but the shift is substantial. How does this relate to later arguments (Fig 6 onwards) made with regard to peptide-MHC that T cells are being positively and negatively selected on, if at all?

Both *Sf*TCR^{mini} and *Sf*C57BL/6 express the same diverse set of endogenous self-peptides, but the former *Sf* strain has also restricted TCR^{mini} repertoire that can delay onset and progression of the disease. On the other hand, *Sf*TCR^{mini}A^bEp mice develop autoimmunity earlier than *Sf*TCR^{mini} mice, regardless that former mice have less diverse TCRs. However, in *Sf*TCR^{mini}A^bEp the original selecting self-peptide is also exclusively expressed in the periphery. Partial replacement of A^bEp complexes on hematopoietic APCs by other self-peptides (A^bEp63K or WT) postpones an onset of autoimmunity (Fig.6). We propose that around 1/3 of CD4Foxp3⁻ cells pass thymic selection by engaging agonist self-peptides but escape deletion. In mice that lack functional Tregs these cells are activated by the same or similar self-peptides in the periphery (also see new SFig15 for the proposed model).

- What is the rationale for showing the increased percent as well as total numbers of CD4 T cells in tissue in *Sf*TCR-mini mice (Fig 1e)? The authors do not explain this. It seems that the latter (cell counts) is enough to make the point (and it would be clearer if this was shown on a log-scale) of greater T cell infiltration of tissues in conjunction with the activation data from Fig1c,d. Following the Referee suggestion, in the revised manuscript only total number of CD4 cells is shown. We also removed graphs of CD4% from Fig. 1 and 4. Originally these graphs were included to provide a statistical means for presented dot plots.

- The authors state that “the proportion of CD4 cells expressing FoxP3-GFP was lower than in healthy mice” in *Sf*TCR-mini mice, but the data shown in Fig 1F,G only substantiates this in the colon. Could the authors please

clarify? As suggested, we corrected the description that only in the colon of *SfTCR^{mini}* mice the proportions of CD4Foxp3⁺ cells were lower (i.e. statistically significant).

Fig. 2

- This figure is not easy to follow due to a lack of details regarding the experiment performed and the data being shown. Specifics to clarify:

o Are the authors assessing the TCR alpha and beta chains on a single-cell level from individual cells? , is the “CDR3 sequence” shown in B the joint alpha-beta pair? If so, how are CDR3 being considered where there are >1 alpha chain in the cell? In *TCR^{mini}* mice all CD4 cells express the same TCR β chain (V β 14D β 2J β 2.6). We sequenced TCR α CDR3 regions (A-from CD4Foxp3⁻ and B from CD4Foxp3⁺ cells as indicated in the heat-map titles). All *TCR^{mini}* mice lack endogenous TCR α loci, and *TCR^{mini}* locus generates a single in-frame TCR α chain. More than one TCR α transcript per cell will interfere with sequencing CDR3 α from single cells that occurs very rarely ⁵.

o The stars in Fig 2A are not explained until the next results section, making this confusing. The authors should consider presenting this aspect in Fig 3 instead. Former Fig.2A is now a Fig. 3D that shows which of 50 dominant TCRs found on *SfCD4Foxp3⁻* T cells and corresponding subset in control mice are confirmed as autoreactive based on an *ex vivo* activation of *SfCD4Foxp3⁻* hybridomas by C57BL/6 DCs.

o How many mice from each strain were sequenced? In the calculation of overlap between strains, how does this compare to overlap between mice? (also for MII index, this is important to understand). What is the increase in overlap as a result of outgrowth or autoreactive clonotypes, rather than just overlap due to reduced repertoire diversity? We sequenced TCRs from various organs from 3 individual *Sf* and 3 control *TCR^{mini}* mice. Three MII values were calculated for each pair of *Sf* or controls, then averaged and cross-compared (Fig. 2C). MII index measures pairwise similarities between populations by considering the overlap and relative abundances of TCRs similarity between aggregates of all samples from each strain. Regardless that *TCR^{mini}* repertoire has reduced diversity, similarity indices calculated for CD4Foxp3⁻ and CD4Foxp3⁺ subsets were around 0.2 ⁵, demonstrating that reduced diversity of *TCR^{mini}* repertoire does not universally increase the similarity between various compared CD4 subsets.

o Given that the goal of this analysis is the comparison of sequence presence/absence and frequency between strains, it would perhaps be more useful to plot the concordance (i.e., *TCRmini* on one axis and *sfTCRmini* on other axis with heat map to indicate the correlation rather than overall frequency). The heatmaps as they stand are difficult to interpret with regard to quantitative patterns observed and the main text does not provide enough explanation. We propose to retain heat maps that concurrently show frequencies of identical TCRs on Foxp3⁺ or Foxp3⁻ subsets in different tissues in *Sf* and healthy controls (each row depicts frequencies of one TCR). The goal of this illustration was to show in which organs/subsets TCRs are shared or unique. Following the Referee 3 suggestion, we now also provide additional plots that shows correlation for these dominant TCRs in both mice (supplemental Fig. S2). Notably, multiple dominant clones that express autoreactive TCRs (i.e. activated *ex vivo* by C57BL/6 DCs), were more prominently represented in *TCR^{mini}* than *SfTCR^{mini}* mice (see Fig. 3D)

o In the statement “we examined the repertoire of $\alpha\beta$ TCRs expressed by autoreactive CD4 T cell clones in *Sf* mice”, what evidence is there that these are “autoreactive”? Is this an inference based on increase in frequency? If so, this should be clarified and explained why this is a valid assumption.

Estimates of autoreactive CD4Foxp3⁻ cells in different Tregs-deficient mice reflect percent of autoreactive hybridomas established from these cells from *SfTCR^{mini}*, *SfC57BL/6*, DT-treated C57BL/6Foxp3^{DTR/GFP} mice (Fig. 3) and *SfTCR^{mini}A^bEp* mice (Fig. 5).

Also, we sequenced TCRs from autoreactive hybridomas representing CD4 cells from *Sf*TCR^{mini} and *Sf*TCR^{mini}A^bEp mice and extrapolated their sequences to corresponding repertoires retrieved from healthy TCR^{mini} strains (to find identical sequences in corresponding HTS dataset of TCR α CDR3 regions from control's CD4Foxp3⁻ cells). Frequencies of all mutual TCRs were added to approximate their abundance in repertoire from healthy mice.

Higher frequencies of particular TCRs in *Sf* mice as compared to healthy mice based on HTS will only provide indirect evidence for these clones clonal expansions. However, proliferating *Sf*CD4 cells include bystander cells dividing in proinflammatory milieu. Our approach is more precise because it includes a direct test of each TCR autoreactivity. Likely, not all autoreactive TCRs from *Sf* strain have been represented on examined hybridomas and some self-reactivities will be difficult to recapitulate *ex vivo*, thus our approximations can be underestimated.

o Fig 2B is cited with reference to a statement about “public TCRs” but this is not data that is included in the figure. In the revised manuscript we changed term “public” was replaced by “dominant” or “abundant”.

- The conclusion that “one third of CD4FoxP3⁻ cells from healthy TCR^{mini} expressed potentially autoreactive abTCRs” is not substantiated by the data. It is not clear where this number comes from, and whether the caveats listed above have been considered in this analysis. As explained above, 1/3 of autoreactive CD4Foxp3⁻ clones was calculated as the proportion of autoreactive *Sf*CD4Foxp3⁻ hybridomas activated by C57BL/6 DCs. For TCR^{mini} mice, this estimate was also confirmed by assessing the percent of CD4Foxp3⁻ cells that share the same TCRs with autoreactive hybridomas.

Figure 3

- In this sentence “analyzed hybridomas have not made IL-2 constitutively and expressed only..” (grammar issues), could the authors clarify whether they mean that in the absence of stimulation the hybridomas do not respond either by IL2 production or by Nur77 induction? Explain that unstimulated made no IL-2 an only none to low nur77^{GFP}. Very rarely hybridomas make IL2 constitutively and these were discarded, but hybridomas can have a different basal expression of Nur77^{GFP} (as also reported for T cells from Nur77^{GFP} reporter mice (Moran et al)). Fig. 3 shows levels of Nur77^{GFP} expression following co-culture with C57BL/6 DCs for individual hybridomas as “fold change over the background”, where increase in Nur77^{GFP} was calculated after dividing by basal reporter expression.

- It would be good to include some raw data examples from the hybridoma experiment that is summarized in A, both with regard to the IL2 response and the Nur77 upregulation. Include examples for Nur77 but for IL-2 it is shown. As suggested, examples of raw data from several hybridomas presented in Fig. 3 are shown in new Fig. S6.

- What was considered background with regard to scoring hybridomas as responsive or unresponsive to autologous APC stimulation? Were the BMDCs/splenocytes activated? At least a 5-fold increase in Nur77^{GFP} expression over the background associated with the secretion of IL-2 was considered a valid response (from *Sf*A^bEp mice). Prior to the use BMDCs were elicited with GMCSF for 6 days and we used freshly isolated splenic DCs.

- Why is the same TCR sequence hybridoma from an *Sf*TCR-mini mouse responding while from a TCR-mini mouse it is not? Is this a result of inhibitory receptors expressed by the cell? Does this not imply that these T cells are fundamentally different, and it is not simply the absence of Tregs which releases the cells from “dormancy”? We propose that the lack of functional Tregs decreases activation threshold of *Sf*CD4Foxp3⁻ cells by self-peptides that is inherited by hybridomas. We found no evidence that expression of inhibitory receptors PD-1 or CTLA-4 correlates with hybridomas unresponsiveness to C57BL/6 DCs. Expression of PD-1 but not CTLA-4 was often elevated by *Sf*CD4Foxp3⁻ cells and established from these cells hybridomas (new Fig.S7). DE genes in *Sf*CD4Foxp3⁻ cells and their counterparts from control mice are shown in Fig. 7. Although not fundamentally different scRNAseq highlighted that more than half of *Sf*CD4Foxp3⁻ cells have features of autoreactive/cytotoxic effectors. New results

shown in Fig. S11 suggest that observed differences in response to C57BL/6 DCs reflect changes in epigenetic regulation of TCR signaling in *Sf*CD4Foxp3⁻ hybridomas.

Fig. 4

- It is unclear why the authors do the experiments presented here. Why would the data shown in Figs 1-3 naturally lead to experiments designed to restrict the TCR repertoire to recognition of a single peptide? Can the authors make more explicit what question is being answered here? Currently it seems more like these mice were in-hand and thus examined, rather than there being a particular hypothesis being tested with regard to the presence of dormant CD4 T cells in the repertoire.

Because 1/3 of *Sf*CD4Foxp3⁻ hybridomas responded to self-peptides presented by C57BL/6 DCs, we wished to test the hypothesis that these autoantigens can represent positively selecting self-peptides. It is known that the majority of self-peptides bound to A^b from thymic epithelial cells overlap with respective peptides eluted from peripheral APCs¹², but in WT mice identities of self-peptide that select particular TCR(s) are unknown. Results from *Sf* "single peptide" mice suggest that a fraction of CD4Foxp3⁻ thymocytes pass positive selection by recognizing Ep self-peptide as agonist. In the periphery, these mature CD4Foxp3⁻ can cause autoimmunity unless stopped by Tregs. Thus, introducing *Sf* mutation to A^bEpTCR^{mini}Foxp3^{GFP} mice was purposely designed to test a question that is difficult to address in WT mice.

- Why would the expression of a single autoantigen "improve the stringency of negative selection" rather than greatly curtail positive selection? In fact, doesn't the accelerated autoimmunity in these mice suggest that negative selection is less stringent?

We stated that the exclusive expression of A^bEp self-complexes enhances negative selection because central tolerance is more effective for abundant than underrepresented self-peptides^{13, 14}

Mice expressing only A^bEp (or other covalently linked "single peptides") are not prone to autoimmunity, have a normal lifespan and contain approximately 1/5 of CD4⁺ T cells found in WT mice. These CD4⁺ T cells also vigorously respond to foreign and natural self-peptides from WT mice, demonstrating that positive selection has not drastically limited their ability to respond to non-self-antigens. We hypothesize that in *Sf*A^bEp mice co-expression of the same self-peptide in the thymus and in the periphery accelerates an onset of autoimmunity.

- What is "rambling autoimmunity"? We corrected this error. Should read "rampant".

- Why is the autoimmunity in *Sf*TCR^{mini}A^bEp mice unexpected, given previous work showing that the repertoire in single-peptide mice generates T cells which are very MHC-reactive? (Lucas and Germain, 1996 for a review).

CD4 cells from A^bEp mice are activated by allo-MHC peptide/complexes and A^b bound with natural self-peptides, but not by self A^bEp complexes³, unless tested cells originate from *Sf*A^bEp mice. As shown in Fig. 5B *Sf*CD4Foxp3⁻ cells are rarely activated by A^b stabilized with another single peptide.

- Also see Fig1 comments with regard to some of the data shown here that applies to this figure as well. We changed survival curves graphics and duration to weeks. We also compared the total number of CD4 cells and removed plots and diagrams referring to these cell percentages.

Figure 5/6

- The hypothesis presented (that more autoreactive CD4 cells mature and require continual suppression by Tregs if selected positively and negatively on the same peptide-MHC complex) would be more compelling if the data clearly showed that the characterized autoreactive cells are being suppressed by T cells, rather than pushed into autoreactivity by the inflammatory milieu and tissue damage in Treg-deficient mice. The authors should also explain why this particular hypothesis is proposed. What are possible alternatives? In adoptive transfer

experiments, there is no preexisting inflammatory milieu and autoreactive cells continue to incite autoimmunity only in A^bEpTCR α - recipients, which can be prevented by Tregs-co-transfer (Fig.6G). A similar outcome was reproduced when naïve CD4Foxp3⁻ cells from healthy TCR^{mini}A^bEp mice were transferred to lymphopenic A^bEpTCR α - mice unless these cells were suppressed by co-transfer of Tregs.

We show that onset and progression of the disease after adoptive transfer to lymphopenic mice or in bone marrow chimeras is accelerated when positively selecting peptide matches self-peptide presented by peripheral APCs. Thus, when Tregs are dysfunctional (or ablated), deletion on selecting peptide does not suffice to prevent autoreactive responses to the same or similar peptide(s) in the periphery.

We are unsure what the Reviewer had in mind proposing alternative hypotheses.

- Labeling in the figure to make clear which bone marrow chimeras result in which outcome with regard to positive and negative selection on what cells would greatly aid in the clarity of figure 6. To improve this Figure clarity, we have added descriptions of selecting ligands to figure legend.

Figure 7

- Clusters 1-4 should be labeled in Fig 7A. We added additional clusters labels to this Figure.

- By limiting the comparison here to activated T cells in the TCR-mini strain, are the authors not missing the majority of truly 'dormant' (still naïve) cells? Would the majority of the gene signature differences in this comparison not simply be a result of the comparison of true effector cells actively responding to peptide-MHC with quiescent, memory-phenotype cells generated for instance by neonatal lymphopenia-induced proliferation?

Comparison of transcriptional profiles of activated Sf effectors to naïve WT cells will surely generate a lot of DE genes of which most will reflect these cells different activation status, as already shown for naïve and activated cells from WT mice. Therefore, we choose to compare DE genes between activated cells from Sf and WT strains. We agree with the Referee that the fraction of naïve CD4 in WT mice has autoreactive potential.

- What is the y-axis in Fig 7C? fold change? CPM? Y-axis in Fig. 7C shows log normalized count (also see Methods)

- Fig 7F, which way is the data plotted with regard to fold change (right side are sfTCRmini or TCRmini upregulated genes)? The Referee is correct. Right side shows genes upregulated in SfTCR^{mini} and left in control TCR^{mini}.

- How many genes are different between cells with shared TCRs from the two strains ? The conclusion that cells are autoreactive in the sfTCR-mini compared to the TCRmini based on the extensive data collection at the single-cell level is underwhelming. What did the sc-RNA level analyses reveal beyond that the cells in sfTCR are actively responding while those in the TCRmini strain are not? These analyses seem primarily confirmatory rather than revealing anything the authors did not already know.

614 genes were DE between CD4Foxp3⁻ cells expressing shared TCRs from Sf and control mice, of which 36 genes had 2-fold or more change in expression level which was statistically significant.

scRNAseq revealed that more than half of Sf cells which shared TCRs with corresponding cells from WT mice grouped in Cluster 1. This cluster consists of cells with high expression of cytotoxic molecules (granzymes, perforins) that can cause tissue damage and transcription factors (Eomes, Bhlhe40) that limit Foxp3 induction and control cytokines production upon autoimmune inflammation. Corresponding clones that expressed mutual TCRs from WT mice were in Clusters 3 and 4. These results show that identical CD4 clones which in Tregs-sufficient mice appear activated but remain quiescent, in Sf mice express gene profiles compatible with cytotoxic and pro-inflammatory effectors.

Discussion: there is an entire paragraph on PD1 expression but no data on this was included in the main manuscript. Thus, none of the unpublished results cited here can be evaluated. The authors should focus their discussion on data that is shown in the manuscript. To address this concern, we now present new data on PD-1 expression on responding *ex vivo* autoreactive CD4Foxp3⁻ hybridomas (Fig. S7).

A summary model schematic would be very helpful. New Fig. S15 presents current and our proposed models.

Minor comments:

There are quite a few grammar issues/typos that should be corrected (some examples listed here from the start of the manuscript, but by far not exhaustive as there are too many):

- Intro: para 1, Line 8 “although for these cells in vivo pathogenicity has not been examined”
- Intro: para 1, last line “but that global identification..”
- Intro: para2, line 1: grammar “and the presence of tregs which continuously suppress these cells.”
- Intro: para 2, line 7: Fatal “outbreaks” of autoimmunity? This is surely the wrong word.
- Intro: last para “transpire to the peripheral repertoire”. What is meant here?
- Intro: last para, final sentence “dynamically modulate the activation threshold”
- Results section #2, first paragraph, line 2: “This is because inflammation” typo in mice.

Diphtheria toxin is has been used for selective ablation of specific cells, including targeted removal of Tregs in healthy B6 mice that expressed a diphtheria toxin (DT) receptor under Foxp3 promoter (Kim 2007).

We apologize for these errors and in the revised manuscript these and other typos were corrected.

Reference List

1. Su Ih, *et al.* Polycomb Group Protein Ezh2 Controls Actin Polymerization and Cell Signaling. *Cell* **121**, 425-436 (2005).
2. Dobenecker M-W, *et al.* Signaling function of PRC2 is essential for TCR-driven T cell responses. *The Journal of Experimental Medicine* **215**, 1101-1113 (2018).
3. Ignatowicz L, Kappler J, Marrack P. The repertoire of T cells shaped by a single MHC/peptide ligand. *Cell* **84**, 521-529 (1996).
4. Wojciech L, *et al.* The same self-peptide selects conventional and regulatory CD4(+) T cells with identical antigen receptors. *Nat Commun* **5**, 5061 (2014).
5. Pacholczyk R, Ignatowicz H, Kraj P, Ignatowicz L. Origin and T cell receptor diversity of Foxp3+CD4+CD25+ T cells. *Immunity* **25**, 249-259 (2006).
6. Moran AE, *et al.* T cell receptor signal strength in Treg and iNKT cell development demonstrated by a novel fluorescent reporter mouse. *The Journal of Experimental Medicine* **208**, 1279-1289 (2011).

7. Kim JM, Rasmussen JP, Rudensky AY. Regulatory T cells prevent catastrophic autoimmunity throughout the lifespan of mice. *Nat Immunol* **8**, 191-197 (2007).
8. Yi J, *et al.* Unregulated antigen-presenting cell activation by T cells breaks self tolerance. *Proc Natl Acad Sci U S A* **116**, 1007-1016 (2019).
9. Legoux FP, *et al.* CD4⁺ T Cell Tolerance to Tissue-Restricted Self Antigens Is Mediated by Antigen-Specific Regulatory T Cells Rather Than Deletion. *Immunity* **43**, 896-908 (2015).
10. Madi A, *et al.* T-cell receptor repertoires share a restricted set of public and abundant CDR3 sequences that are associated with self-related immunity. *Genome Research* **24**, 1603-1612 (2014).
11. Vrisekoop N, Monteiro JP, Mandl JN, Germain RN. Revisiting Thymic Positive Selection and the Mature T Cell Repertoire for Antigen. *Immunity* **41**, 181-190 (2014).
12. Marrack P, Ignatowicz L, Kappler JW, Boymel J, Freed JH. Comparison of peptides bound to spleen and thymus class II. *The Journal of experimental medicine* **178**, 2173-2183 (1993).
13. Malhotra D, *et al.* Specific patterns of self-antigen expression determine the mechanisms by which polyclonal self-reactive CD4⁺ T cells are tolerized. *The Journal of Immunology* **196**, 55.24-55.24 (2016).
14. Malhotra D, *et al.* Tolerance is established in polyclonal CD4(+) T cells by distinct mechanisms, according to self-peptide expression patterns. *Nature Immunology* **17**, 187-195 (2016).

REVIEWERS' COMMENTS:

Reviewer #1 (Remarks to the Author):

The authors have adequately addressed all comments.

Reviewer #2 (Remarks to the Author):

The authors addressed most points raised by the 3 referees. In particular, the authors made substantial efforts to clarify the manuscript. In addition, they provided more evidences concerning one of the critical point of the study namely the differential ability of self-reactive CD4 T cells, in WT or Scruddy mice, to respond to self-peptides despite sharing TCR sequences. These findings will provide a framework for future studies in the field trying to understand the molecular basis of such differential responsiveness.

Reviewer #3 (Remarks to the Author):

The flow of the manuscript is much better, with a clear narrative thread from one figure to the next and substantial corrections made to the grammar that were affecting the clarity. Text in introduction and results has been re-organized to introduce the model and key concepts from the previous literature. Figures have greatly improved with regard to clarity with all this reviewer's suggestions incorporated. The model figure is a very nice addition and it should be considered putting it into the main figures at the end rather than as one of 15 supplementary figures.

Other points:

- From the previous review, the term "dormant clones" that is used throughout the manuscript is still not explicitly defined in the introduction. The authors state that this is provided in the last sentence of the second paragraph, but this is not the case. The word "dormant clone" does not get used in the introduction at all.
- 15 supplementary figures seems excessive. Can this be reduced to incorporate the essentials only.

**Center for Translational Immunology
Institute for Biomedical Sciences**

*Leszek Ignatowicz, PhD
Professor*

September 20, 2019

NCOMMS-19-14032

Dear Dr. Bondar,

Please find below our response to Referee's comments.

ALL REVIEWERS' COMMENTS:

Reviewer #1 (Remarks to the Author):

The authors have adequately addressed all comments.

We thank Ref 1 for all comments.

Reviewer #2 (Remarks to the Author):

The authors addressed most points raised by the 3 referees. In particular, the authors made substantial efforts to clarify the manuscript. In addition, they provided more evidence concerning one of the critical point of the study namely the differential ability of self-reactive CD4 T cells, in WT or Scruffy mice, to respond to self-peptides despite sharing TCR sequences. These findings will provide a framework for future studies in the field trying to understand the molecular basis of such differential responsiveness.

We thank Ref 2 for all comments.

Reviewer #3 (Remarks to the Author):

The flow of the manuscript is much better, with a clear narrative thread from one figure to the next and substantial corrections made to the grammar that were affecting the clarity. Text in introduction and results has been re-organized to introduce the model and key concepts from the previous literature. Figures have greatly improved with regard to clarity with all this reviewer's suggestions incorporated. The model figure is a very nice addition and it should be considered putting it into the main figures at the end rather than as one of 15 supplementary figures.

We concur with Ref 3 and request to reassign a model figure originally shown as Supplemental figure 15 to be included as Figure 8 in main text.

Other points:

- From the previous review, the term "dormant clones" that is used throughout the manuscript is still not explicitly defined in the introduction. The authors state that this is provided in the last sentence of the second

Georgia State University
Research Science Center, Room 314
100 Piedmont Avenue
Atlanta, Georgia 30303

Tel: 404-413-6685
Fax: 404-413-3580
E-mail: lignatowicz@gsu.edu

paragraph, but this is not the case. The word “dormant clone” does not get used in the introduction at all.

In the revised abstract there is a statement that in our opinion explains that by dormant, potentially pathogenic cells we consider CD4⁺Foxp3⁻ T cells normally suppressed by CD4⁺Foxp3⁻ Tregs.

“Here we compare immune repertoires of Treg-deficient and Treg-sufficient mice to find Tregs continuously constraining one-third of mature CD4⁺Foxp3⁻ cells from converting to pathogenic effectors in healthy mice. These dormant pathogenic clones...”

• 15 supplementary figures seems excessive. Can this be reduced to incorporate the essentials only.

The number of supplemental Figures doubled because Referees requested additional data to be added to the manuscript. We feel uncomfortable to arbitrarily remove selected Supplemental Figures after these have been submitted and reviewed, and propose to retain all supplemental figures submitted for the revision.

We hope that our revisions will meet your approval.

Regards,

Leszek Ignatowicz